

# QuSpin: a Python package for dynamics and exact diagonalisation of quantum many body systems

## Part I: spin chains

Phillip Weinberg[*] and Marin Bukov

Department of Physics, Boston University, 590 Commonwealth Ave., Boston, MA 02215, USA

* weinbe58@bu.edu

## Abstract

We present a new open-source Python package for exact diagonalisation and quantum dynamics of spin(-photon) chains, called QuSpin, supporting the use of various symmetries in 1-dimension and (imaginary) time evolution for chains up to 32 sites in length. The package is well-suited to study, among others, quantum quenches at finite and infinite times, the Eigenstate Thermalisation hypothesis, many-body localisation and other dynamical phase transitions, periodically-driven (Floquet) systems, adiabatic and counter-diabatic ramps, and spin-photon interactions. Moreover, QuSpin's user-friendly interface can easily be used in combination with other Python packages which makes it amenable to a high-level customisation. We explain how to use QuSpin using four detailed examples: (i) Standard exact diagonalisation of XXZ chain (ii) adiabatic ramping of parameters in the many-body localised XXZ model, (iii) heating in the periodically-driven transverse-field Ising model in a parallel field, and (iv) quantised light-atom interactions: recovering the periodically-driven atom in the semi-classical limit of a static Hamiltonian.

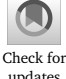

# 1   What problems can I solve with QuSpin?

The study of quantum many-body dynamics comprises a variety of problems, such as dynamical phase transitions (e.g. many-body localisation), thermalising time evolution, adiabatic change of parameters, periodically-driven systems, and many others. In contrast to the tremendous progress made in studying low-energy phenomena based on well-developed sophisticated techniques, such as Quantum Monte Carlo methods [1–3], Density Matrix Renormalisation Group [4,5], Matrix Product States [6], Dynamical Mean-Field Theory [7–9], etc., one of the most popular "cutting-edge" investigation technique for out-of-equilibrium quantum many-body problems remains 'old school' exact diagonalisation (ED).

Over the last years, there have appeared freely accessible numerical packages and libraries which contribute to widespread the use of such numerical techniques among the condensed matter community: the Algorithms and Libraries for Physics Simulations [10–13] (ALPS), C++ libraries for tensor networks: ITensor [14] and Tensor Network Theory Library [15], the Quantum Toolbox in Python (QuTiP) [16,17], as well as the Mathematica Quantum Many-Body Physics Package DiracQ [18] are among the most common available and freely accessible tools. Although some of these are indeed freely accessible, not all of them are considered open-source as they have certain restrictions about the use and/or distribution of the source code. On the other hand, some members of the scientific community have been pushing for more open-source software, notably in the development of Python. The authors benefited greatly from the community development of these powerful numerical tools and so in the same spirit we present in this paper, an optimised open-source Python package for dynamics and exact diagonalisation of quantum many-body spin systems, called QuSpin:

- A major representative feature of QuSpin is the construction of spin Hamiltonians containing arbitrary (possibly non-local in space) many-body operators. One example is the four-spin operator $\mathcal{O} = \sum_j \sigma_j^z \sigma_{j+1}^+ \sigma_{j+2}^- \sigma_{j+3}^z + \text{h.c.}$. Such multi-spin operators are often times generated by the nested commutators typically appearing in higher-order terms of perturbative expansions, such as the Schrieffer-Wolff transformation [19–21] and the inverse-frequency expansion [22,23]. Sometimes they appear in the study of exactly solvable reverse-engineered models.

- Another important feature is the availability to use symmetries which, if present in a given model, give rise to conservation laws leading to selection rules between the many-body states. As a result, the Hilbert space reduces to a tensor product of the Hilbert

spaces corresponding to the underlying symmetry blocks. Consequently, the presence of symmetries effectively diminishes the relevant Hilbert space dimension which, in turn, allows one to study larger systems. Currently, QuSpin supports the following spin chain symmetries:

- total magnetisation (particle number in the case of hard-core bosons)
- parity (i.e. reflection w.r.t. the middle of the chain)
- spin inversion (on the entire chain but also individually for sublattices $A$ and $B$)
- the joint application of parity and spin inversion (present e.g. when studying staggered or linear external potentials)
- translation symmetry
- all physically meaningful combinations of the above

We shall see in Sec. 2, constructing Hamiltonians with given symmetries is done by specifying the desired combination of symmetry blocks.

- As of present date, ED methods represent one of the most reliable ways to safely study the dynamics of a generic quantum many-body system. In this respect, it is important to emphasise that with QuSpin the user can build arbitrary time-dependent Hamiltonians. The package contains built-in routines to calculate the real (and imaginary) time evolution of any quantum state under a user-defined time-dependent Hamiltonian based on SciPy's integration tool for ordinary differential equations [24].

- Besides spin chains, QuSpin also allows the user to couple an arbitrary interacting spin chain to a single photon mode (i.e. quantum harmonic oscillator). In this case, the total magnetisation symmetry is replaced by the combined total photon and spin number conservation. Such an example is discussed in Sec. 2.4.

- Last but not least, QuSpin has been especially designed to construct particularly short and efficient ED codes (typically less than 200 lines, as we explicitly demonstrate in Sec. 2 and App. C). This greatly reduces the amount of time required to start a new study; it also allows users with little-to-no programming experience to do state-of-the-art ED calculations.

Examples of 'hot' problems that can be studied with the help of QuSpin include:

* quantum quenches and quantum dynamics at finite and infinite times

* adiabatic and counter-diabatic ramps

* periodically driven (Floquet) systems

* many-body localisation, Eigenstate Thermalisation hypothesis

* quantum information

* quantised spin-photon interactions and similar cavity QED related models

* dynamical phase transitions and critical phenomena

* machine learning with quantum many-body systems

This list is far from being complete, but it can serve as a useful guideline to the interested user.

Overall, we believe QuSpin to be of particular interest to both students and senior researchers, who can use it to quickly test new exciting ideas, and build up intuition about quantum many-body problems.

## 2 How do I use QuSpin?

One of the main advantages of QuSpin is its user-friendly interface. To demonstrate how the package works, we shall guide the reader step by step through a short snippets of Python code. In case the reader is unfamiliar with Python, we kindly invite them to accept the challenge of learning the Python basics, while enjoying the study of quantum many-body dynamics, see App. B.

*Installing QuSpin is quick and efficient; just follow the steps outlined in App. A.*

Below, we stick to the following general guidelines: first, we define the problem containing the physical quantities of interest and show their behaviour in a few figures. After that, we present the QuSpin code used to generate them, broken up into its building blocks. We explain each step in great detail. The complete code, including the lines used to generate the figures shown below, is available in App. C. It is not our purpose in this paper to discuss in detail the interesting underlying physics of these systems; instead, we focus on setting up the Python code to study them with the help of QuSpin, and leave the interested reader figure out the physics details themselves.

### 2.1 Exact diagonalisation of spin Hamiltonians

*Physics Setup*—Before we show how QuSpin can be used to solve more sophisticated time-dependent problems, let us discuss how to set up and diagonalise static spin chain Hamiltonians. We focus here on the XXZ model in an magnetic field

$$H = \sum_{j=0}^{L-2} \frac{J_{xy}}{2} \left( S_{j+1}^+ S_j^- + \text{h.c.} \right) + J_{zz} S_{j+1}^z S_j^z + h_z \sum_{j=0}^{L-1} S_j^z, \tag{1}$$

where $J_{xy}$ and $J_{zz}$ are the $xy$- and $zz$-interaction strengths, respectively, and $h_z$ is the external field along the $z$ direction. Note that we enumerate the $L$ sites of the chain by $j = 0, 1, \ldots, L-1$ to conform with Python's array indexing convention. We shall assume open boundary conditions.

*Code Analysis*—Let us now build and diagonalise $H$ using QuSpin. First, we load the required Python packages. Note that we adopt the commonly used abbreviation for NumPy, `np`.

```python
from quspin.operators import hamiltonian # Hamiltonians and operators
from quspin.basis import spin_basis_1d # Hilbert space spin basis
import numpy as np # generic math functions
```

Next, we define the physical model parameters. In doing so, it is advisable to use the floating point when the coupling is meant to be a non-integer real number, in order to avoid problems with division: for example, 1 is the integer 1 while `1.0` – the corresponding float. For instance, in Python 2.7, we have `0.5≠1/2=0`, but rather `0.5=1.0/2.0`.

```python
##### define model parameters #####
L=12 # system size
Jxy=np.sqrt(2.0) # xy interaction
Jzz_0=1.0 # zz interaction
hz=1.0/np.sqrt(3.0) # z external field
```

To set up any Hamiltonian, we need to calculate the basis of the Hilbert space it is defined on, see `line 13` below. Note that, since we work with spin operators here, it is required to pass the flag `pauli=False`; failure to do so will result in a Hamiltonian defined in terms of

the Pauli spin matrices $\sigma_i^\mu = 2S_i^\mu$. One can also display the basis using the command `print basis`.

```
11 ##### set up Heisenberg Hamiltonian in an enternal z-field #####
12 # compute spin-1/2 basis
13 basis = spin_basis_1d(L,pauli=False)
```

The XXZ Hamiltonian $H$ from Eq. (1) obeys certain symmetries. In particular, one can specify a magnetisation sector (a.k.a. filling) using the `basis` optional argument `Nup=int`, where `int`$\in [0, L]$ is any integer to specify the number of up-spins, see `line 14`. However, magnetisation is not the only integral of motion – the model also conserves parity, i.e. reflection w.r.t. the middle of the chain. The parity operator has eigenvalues $\pm 1$ and thus further divides the Hilbert space into two new subspaces. To restrict the Hamiltonian to either one of them, we use the `basis` optional argument `pblock=`$\pm 1$. Since parity and magnetisation commute, it is also possible to request them simultaneously, see `line 15`. We stress that each one of the `lines 13-15` is sufficient to build the `basis` on its own and we only show them all here for clarity.

```
14 basis = spin_basis_1d(L,pauli=False,Nup=L//2) # zero magnetisation sector
15 basis = spin_basis_1d(L,pauli=False,Nup=L//2,pblock=1) # and positive parity
      sector
```

In QuSpin, many-body operators $JS_{i_1}^{\mu_1} \ldots S_{i_n}^{\mu_n}$ are defined by a string of letters $\mu_1, \ldots \mu_n$, representing the operator types, $\mu_i \in$ ["+","-","x","y","z"], together with a site-coupling list $[J, i_1, \ldots, i_n]$ which holds the coupling and the indices for the sites $i$ that each spin operator acts at on the lattice. Setting up the spin-spin operators in the XXZ model goes as follows. First, we need to define the site-coupling lists `J_zz`, `J_xy` and `h_z`. To uniquely specify a two-spin interaction, we need (i) the coupling, and (ii) – the labels of the sites the two operators act on. QuSpin uses Python's indexing convention meaning that the first lattice site is always $i = 0$, and the last one: $i = L - 1$. For example, for the $zz$-interaction, the coupling is `Jzz`, while the two sites are the nearest neighbours `i,i+1`. Hence, the list `[Jzz,i,i+1]` defines the bond operator $J_{zz}(0)S_i^\mu S_{i+1}^\nu$ (we specify $\mu$ and $\nu$ in the next step). To define the total interaction energy $J_{zz}(0)\sum_{i=0}^{L-2} S_i^\mu S_{i+1}^\nu$, all we need is to loop over the $L-2$ bonds of the open chain[12]. In the same spirit one can define boundary or single-site operators, such as `h_z`. It is also possible to set up multi-spin operators, as we show in Sec. 2.3.

```
16 # define operators with OBC using site-coupling lists
17 J_zz = [[Jzz_0,i,i+1] for i in range(L-1)] # OBC
18 J_xy = [[Jxy/2.0,i,i+1] for i in range(L-1)] # OBC
19 h_z=[[hz,i] for i in range(L-1)]
```

The above lines of code specify the coupling but not yet which spin operators are being coupled. i.e. we have not yet fixed $\mu$ and $\nu$. To do this, we need to create a `static` and/or `dynamic` operator list. As the name suggests, static lists define time-independent operators. Given the site-coupling list `J_xy` from above, it is easy to define the operator $J_{xy}/2\sum_{i=0}^{L-2} S_i^+ S_{i+1}^-$ by specifying the spin operator type in the same order as the site indices appear in the corresponding site-coupling list: `[["+-",J_xy]]`. In other words, the order `"+-"` corresponds directly to the site-index order `"i,i+1"`. Similarly, one should set up the hermitian conjugate term $J_{xy}/2\sum_{i=0}^{L-2} S_i^- S_{i+1}^+$ as `[["-+",J_xy]]`. In the end, one can concatenate these operator lists to produce the static part of the Hamiltonian.

```
20 # static and dynamic lists
21 static = [["+-",J_xy],["-+",J_xy],["zz",J_zz]]
```

---

[1]The Python expression `range(L-1)` produces all integers between `0` and `L-2` including.

[2]For periodic boundary conditions we need a connection from `L-1` to `0`, which is easily accomplished with the modulo (%) operator and looping over all sites: `[[J_zz_0,i,(i+1)% L] for i in range(L)]`.

If the Hamiltonian has time dependence, it is defined using `dynamic` lists. Since we are dealing with a static problem in this section, we set the `dynamic` to an empty list. In the following three sections, we show how to set up non-trivial time-dependent Hamiltonians.

```
22  dynamic=[]
```

Once the static and dynamic lists are set up, building up the corresponding Hamiltonian is a one-liner. In QuSpin, this is done using the `hamiltonian` constructor class, see `line 24` below. The first required argument is the `static` list, while the second one – the `dynamic` list. These two arguments must appear in this order. Another argument is the `basis`, which carries the necessary information about symmetries. Yet whether a given Hamiltonian has these symmetries, depends on the operators defined in the static and dynamic lists. The `hamiltonian` class performs an automatic check on the Hamiltonian for hermiticity, the presence of magnetisation conservation and other requested symmetries, and raises an error if these checks fail.

```
23  # compute the time-dependent Heisenberg Hamiltonian
24  H_XXZ = hamiltonian(static,dynamic,basis=basis,dtype=np.float64)
```

Having set up the Hamiltonian, we now briefly discuss a few ED routines. If one is only interested in the spectrum E, one can obtain it as follows:

```
26  ##### various exact diagonalisation routines #####
27  # calculate entire spectrum only
28  E=H_XXZ.eigvalsh()
```

If, on top, one also needs the unitary matrix `V` with the corresponding eigenvectors in its columns, the proper command is

```
29  # calculate full eigensystem
30  E,V=H_XXZ.eigh()
```

Often times, one does not need to fully diagonalise $H$, but a part of the spectrum suffices. For instance, if one is interested in the many-body bandwidth of the model, it can be computed from the smallest and largest eigenvalues. This can be done efficiently using the `eigsh` attribute (**eig**envalues of a **s**parse **h**ermitian matrix), see `line 32` below. The optional argument `k=2` ensures that only two eigenstates are calculated. To determine which ones, the argument `which="BE"` specifies them to be the two states at Both Ends of the spectrum[3]. Convergence of the underlying diagonalisation algorithm is enforced by explicitly specifying the number of maximal iterations: `maxiter=1E4`. If we do not want the eigenstates returned, we use `return_eigenvectors=False`.

```
31  # calculate minimum and maximum energy only
32  Emin,Emax=H_XXZ.eigsh(k=2,which="BE",maxiter=1E4,return_eigenvectors=False)
```

Last, we show how to find that eigenenergy and eigenstate, closest to a given predefined energy `E_star`. This is also done using the `eigsh` attribute. Since we request only one state, we set `k=1`. The predefined energy is then passed using the optional argument `sigma`[4]. More on how `eigsh` works can be found in the SciPy online documentation.

```
33  # calculate the eigenstate closest to energy E_star
34  E_star = 0.0
35  E,psi_0=H_XXZ.eigsh(k=1,sigma=E_star,maxiter=1E4)
36  psi_0=psi_0.reshape((-1,))
```

The entire code is available in Example Code 1.

---

[3]This option is currently available only for *real* Hamiltonians

[4]If `sigma` falls exactly on an eigenvalue of the matrix (within machine precision) this function will stop the execution of the program and display an error.

## 2.2 Adiabatic control of parameters in many-body localised phases

*Physics Setup*—Strongly disordered many-body models have recently enjoyed a lot of attention in the theoretical condensed matter community. It has been shown that, beyond a critical disorder strength, these models undergo a dynamical phase transition from an delocalised ergodic (thermalising) phase to a many-body localised (MBL), i.e. non-conducting, non-thermalising phase, in which the system violates the Eigenstate Thermalisation hypothesis [25–31].

In our first QuSpin example, we show how one can study the adiabatic control of model parameters in many-body localised phases. It was recently argued that the adiabatic theorem does not apply to disordered systems [32]. On the other hand, controlling the system parameters in MBL phases is of crucial experimental [33–37] significance. Thus, the question as to whether there exists an adiabatic window for some, possibly intermediate, ramp speeds (as is the case for periodically-driven systems [38]), is of particular and increasing importance.

Let us consider the XXZ open chain in a disordered $z$-field with the time-dependent Hamiltonian

$$H(t) = \sum_{j=0}^{L-2} \frac{J_{xy}}{2} \left( S_{j+1}^+ S_j^- + \text{h.c.} \right) + J_{zz}(t) S_{j+1}^z S_j^z + \sum_{j=0}^{L-1} h_j S_j^z,$$

$$J_{zz}(t) = (1/2 + vt) J_{zz}(0), \tag{2}$$

where $J_{xy}$ is the spin-spin interaction in the $xy$-plane, disorder is modelled by a uniformly distributed random field $h_j \in [-h_0, h_0]$ of strength $h_0$ along the $z$-direction, and the spin-spin interaction along the $z$-direction – $J_{zz}(t)$ – is the adiabatically-modulated (ramp) parameter. In the following, we set $J_{zz}(0) = 1$ as the energy unit. It has been demonstrated that this model exhibits a transition to an MBL phase [39]. In particular, for $h_0 = h_{\text{MBL}} = 3.72$ the system is in a many-body localised phase, while for $h_0 = h_{\text{ETH}} = 0.1$ the system is in the ergodic (ETH) delocalised phase. We now choose the ramp protocol $J_{zz}(t) = (1/2 + vt) J_{zz}(0)$ with the ramp speed $v$, and evolve the system with the Hamiltonian $H(t)$ from $t_i = 0$ to[5] $t_f = (2v)^{-1}$. We choose the initial state $|\psi_i\rangle = |\psi(t_i)\rangle$ from the middle of the spectrum of $H(t_i)$ to ensure typicality; more specifically we choose $|\psi_i\rangle$ to be that eigenstate of $H(t_i)$ whose energy is closest to the middle of the spectrum of $H(t_i)$, where the density of states, and thus the thermodynamic entropy, is largest.

To determine whether the system can adiabatically follow the ramp, we use two different indicators: (i) we evolve the state up to time $t_f$ and project it onto the eigenstates of $H(t_f)$. The corresponding diagonal entropy density:

$$s_d = -\frac{1}{L} \text{tr}[\rho_d \log \rho_d], \qquad \rho_d = \sum_n |\langle n|\psi(t_f)\rangle|^2 |n_f\rangle\langle n_f|, \tag{3}$$

in the basis $\{|n_f\rangle\}$ of $H(t_f)$ at small enough ramp speeds $v$, is a measure of the delocalisation of the time-evolved state $\psi(t_f)\rangle$ among the energy eigenstates of $H(t_f)$. If, for instance, after the ramp the system still occupies a single eigenstate $|\tilde{n}_f\rangle$, then $s_d = 0$.

The second measure of adiabaticity we use is (ii) the entanglement entropy density

$$s_{\text{ent}}(t_f) = -\frac{1}{|A|} \text{tr}_A \left[ \rho_A(t_f) \log \rho_A(t_f) \right], \qquad \rho_A(t_f) = \text{tr}_{A^c} |\psi(t_f)\rangle\langle\psi(t_f)|, \tag{4}$$

of subsystem A, defined to contain the left half of the chain and $|A| = L/2$. We denoted the reduced density matrix of subsystem A by $\rho_A$, and $A^c$ is the complement of A.

Figure 1 shows the entropies vs. ramp speed in the MBL and ETH phases. The interesting underlying physics is, however, beyond the purpose of this paper.

---

[5]Notice that $t_f \to \infty$ as $v \to 0$ and thus, the total evolution time increases with decreasing the ramp speed $v$.

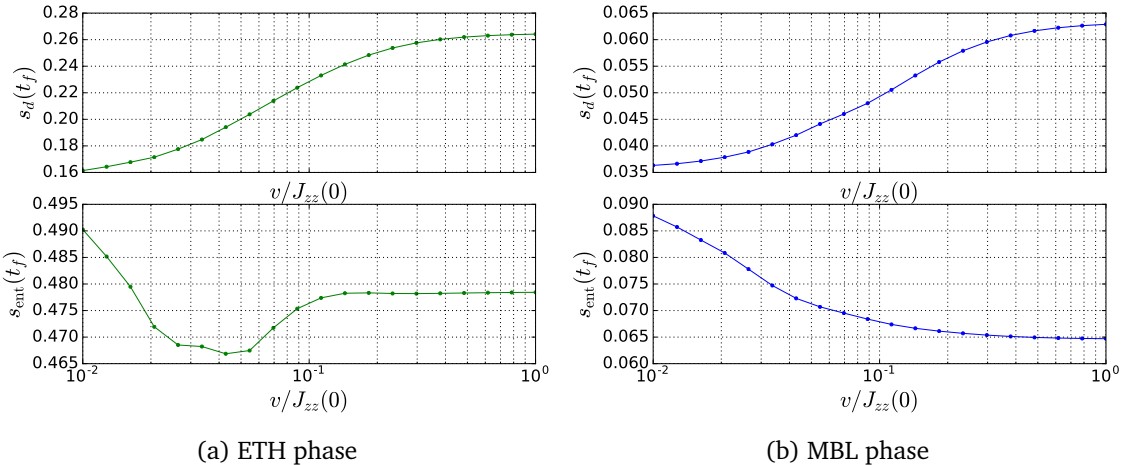

(a) ETH phase                    (b) MBL phase

Figure 1: Diagonal and entanglement entropy densities as a function of the ramp speed in the MBL and delocalised (ETH) phases of the ramped disordered XXZ model. The ramped protocol is chosen as $J_{zz}(t) = (1/2 + vt)J_{zz}(0)$. The parameters are $J_{xy}/J_{zz}(0) = 1.0$, $h_{\mathrm{MBL}}/J_{zz}(0) = 3.9$, $h_{\mathrm{ETH}}/J_{zz}(0) = 0.1$, and $L = 10$. Disorder averaging was performed over 1000 realisations.

*Code Analysis*—Let us now explain how one can study this problem numerically using QuSpin. We begin by first loading the required Python packages.

```
1  from quspin.operators import hamiltonian # Hamiltonians and operators
2  from quspin.basis import spin_basis_1d # Hilbert space spin basis
3  from quspin.tools.measurements import ent_entropy, diag_ensemble # entropies
4  from numpy.random import ranf,seed # pseudo random numbers
5  from joblib import delayed,Parallel # parallelisation
6  import numpy as np # generic math functions
7  from time import time # timing package
```

Since we want to produce many realisations of the data and average over disorder, we specify the simulations parameters: `n_real` is the number of disorder realisations, while `n_jobs` is the `joblib` parallelisation parameter which determines how many Python processes to run simultaneously[6].

```
9  ##### define simulation parameters #####
10 n_real=20 # number of disorder realisations
11 n_jobs=2 # number of spawned processes used for parallelisation
```

Next, we define the physical model parameters.

```
13 ##### define model parameters #####
14 L=10 # system size
15 Jxy=1.0 # xy interaction
16 Jzz_0=1.0 # zz interaction at time t=0
17 h_MBL=3.9 # MBL disorder strength
18 h_ETH=0.1 # delocalised disorder strength
19 vs=np.logspace(-2.0,0.0,num=20,base=10) # log_2-spaced vector of ramp speeds
```

The time-dependent disordered Hamiltonian consists of two parts: the time-dependent XXZ model which is disorder-free, and the disorder field whose values differ from one realisation to another. We focus on the XXZ part first. Let us code up the driving protocol given by $J_{zz}(t) = (1/2 + vt)J_{zz}(0)$. As already explained, our goal is to obtain the disorder-averaged

---

[6]While one can spawn as many processes as one desires, it is optimal to spawn only about as many processes as there are available cores in the processor.

entropies as a function of the ramp speed $v$. Hence, for each disorder realisation, we need to evolve the initial state many times, each corresponding to a different ramp speed. However, calculating the Hamiltonian every single time is not particularly efficient from the point of view of simulation runtime. We thus want to set up a family of Hamiltonians $\{v : H(t;v)\}$ at once, and we shall employ Python's features to do so. This will require that the drive speed $v$ is *not* a parameter of the function `ramp`, see `line 29`, but is declared beforehand as a global variable. Once, `ramp` has been defined, reassigning $v$ dynamically induces the corresponding change in `ramp` without any need to re-define `ramp` itself. We shall comment on how this works later on in the code.

```
26  ##### set up Heisenberg Hamiltonian with linearly varying zz-interaction #####
27  # define linear ramp function
28  v = 1.0 # declare ramp speed variable
29  def ramp(t):
30      return (0.5 + v*t)
31  ramp_args=[]
```

To set up the static part of the Hamiltonian, we follow the same steps as in Sec. 2.1. Since the Hamiltonian (2) conserves the total magnetisation, the overlap betweens states of different magnetisation sectors vanishes trivially, and we can reach larger system sizes by working in a fixed magnetisation sector. A natural choice is the zero-magnetisation sector which contains the ground state. Parity is broken by the disorder field, so we leave it out.

The time-dependent part of the Hamiltonian is defined using `dynamic` lists. Similar to their static counterparts, one needs to define an operator string, say `"zz"` to declare the specific operator from a site-coupling list. Apart from the site-coupling list `J_zz`, however, a dynamic list also requires a time-dependent function and its arguments, see `line 34` below[7]. In the linearly driven XXZ-Hamiltonian we are setting up here, the function arguments `ramp_args` is an empty list, see `line 31` above. The careful reader might have noticed that there is a certain freedom in coding the coupling of the time-dependent term, $J_{zz}(t) = (1/2 + vt)J_{zz}(0)$: here we choose to include the constant `Jzz_0` in the `zz` site-coupling list and hence this factor is absent in the definition of the `ramp` function. Building the Hamiltonian is straightforward, as we explained in Sec. 2.1.

```
27  # compute basis in the 0-total magnetisation sector (requires L even)
28  basis = spin_basis_1d(L,Nup=L//2,pauli=False)
29  # define operators with OBC using site-coupling lists
30  J_zz = [[Jzz_0,i,i+1] for i in range(L-1)] # OBC
31  J_xy = [[Jxy/2.0,i,i+1] for i in range(L-1)] # OBC
32  # static and dynamic lists
33  static = [["+-",J_xy],["-+",J_xy]]
34  dynamic =[["zz",J_zz,ramp,ramp_args]]
35  # compute the time-dependent Heisenberg Hamiltonian
36  H_XXZ = hamiltonian(static,dynamic,basis=basis,dtype=np.float64)
```

To produce the entropies vs. ramp speed data over many disorder realisations, we define the function `realization` which returns a two NumPy arrays for the MBL and ETH phases respectively. Each array contains values of the values of the diagonal entropy density $s_d$, and the values of the entanglement entropy density $s_{ent}$ for each velocity, as columns of the array. We now walk the reader step by step through the definition of `realization`. The first argument is the vector of ramp speeds, `vs`, required for the dynamics. The second argument is the time-dependent XXZ Hamiltonian `H_XXZ` to which we shall be adding a disordered $z$-field for each disorder realisation. The third argument is the spin `basis` which is required to calculate $s_{ent}$. The fourth (last) argument is the realisation number, which is only used to print a message about the duration of the single realisation run.

---

[7]All functions passed in the `dynamic` list are assumed to be defined with time as the first arguement.

```
38  ##### calculate diagonal and entanglement entropies #####
39  def realization(vs,H_XXZ,basis,real):
40      """
41      This function computes the entropies for a single disorder realisation.
42      --- arguments ---
43      vs: vector of ramp speeds
44      H_XXZ: time-dep. Heisenberg Hamiltonian with driven zz-interactions
45      basis: spin_basis_1d object containing the spin basis
46      n_real: number of disorder realisations; used only for timing
47      """
```

We time each realisation simulation, using the package `time`:

```
48      ti = time() # get start time
```

In order to properly be able to use $H_{XXZ}(t; v)$ as a family of Hamiltonians in $v$ (we shall see exactly how this works in a moment), we explicitly declare the variable $v$ global.

```
50      global v # declare ramp speed v a global variable
```

Since the problem involves disorder, we have to generate multiple disorder realisations. In this case, it is recommended to reset the pseudo-random number generator before any random numbers have been drawn. This is because the code spawns multiple python processes to do the disorder realization in parallel. Therefore, if the pseudo-random number generator is seeded before the new processes are spawned, all the parallel jobs will produce the same disorder realizations.

```
52      seed() # the random number needs to be seeded for each parallel process
```

Next, we set up the full disordered time-dependent Hamiltonian of the problem, given by $H(t) = H_{XXZ}(t) + \sum_j h_j S_j^z$. The random field $h_j$ differs from one realisation to another and is, therefore, defined inside the `realization` function. Recall that we want to compare the localised with the delocalised regimes, corresponding to the disorder strengths $h_{MBL}$ and $h_{ETH}$, respectively. For each lattice site $i$ we draw a random number `unscaled_fields[i]` uniformly in the interval $[-1, 1]$, and store it in the vector `unscaled_fields`, see `line 55` below. Building the external $z$-field proceeds in exactly the same way as before: (i) we calculate the site-coupling list, `line 57`, (ii) we designate that the operator is along the $z$-axis by defining a static operator list, `line 59`, and (iii) we use the already computed spin basis to construct the operator matrix with the `hamiltonian` class, `lines 61-62`. QuSpin has the option to disable the default checks on hermiticity, magnetisation (particle number) conservation, and symmetries using the auxiliary dictionary `no_checks` passed straight to `hamiltonian` as keyword arguments. This can allow the user to define non-hermitian operators. Last, in `lines 64-65`, we define the MBL and ETH time-dependent Hamiltonians, corresponding to the two disorder strengths $h_{ETH}$ and $h_{MBL}$.

```
54      # draw random field uniformly from [-1.0,1.0] for each lattice site
55      unscaled_fields=-1+2*ranf((basis.L,))
56      # define z-field operator site-coupling list
57      h_z=[[unscaled_fields[i],i] for i in range(basis.L)]
58      # static list
59      disorder_field = [["z",h_z]]
60      # compute disordered z-field Hamiltonian
61      no_checks={"check_herm":False,"check_pcon":False,"check_symm":False}
62      Hz=hamiltonian(disorder_field,[],basis=basis,dtype=np.float64,**no_checks)
63      # compute the MBL and ETH Hamiltonians for the same disorder realisation
64      H_MBL=H_XXZ+h_MBL*Hz
65      H_ETH=H_XXZ+h_ETH*Hz
```

Let us first focus on the MBL phase. We want the initial state to be as close as possible to an infinite-temperature state within the given symmetry sector. To this end, we can first calculate the minimum and maximum energy, `Emin` and `Emax` of the spectrum of $H_{\text{MBL}}(t=0)$. Then, by taking the 'centre-of-mass' we obtain a number, `E_inf_temp`, which represents the infinite-temperature energy up to finite-size effects, line 72. Note that the `**eigsh_args` is a standard Python way of reading off the arguments by name from a dictionary.

```
67    ### ramp in MBL phase ###
68    v=1.0 # reset ramp speed to unity
69    # calculate the energy at infinite temperature for initial MBL Hamiltonian
70    eigsh_args={"k":2,"which":"BE","maxiter":1E4,"return_eigenvectors":False}
71    Emin,Emax=H_MBL.eigsh(time=0.0,**eigsh_args)
72    E_inf_temp=(Emax+Emin)/2.0
```

The initial state `psi_0` is then that eigenstate of $H_{\text{MBL}}(t=0)$, whose energy is closest to `E_inf_temp`, using optional argument `sigma=E_inf_temp`.

```
73    # calculate nearest eigenstate to energy at infinite temperature
74    E,psi_0=H_MBL.eigsh(time=0.0,k=1,sigma=E_inf_temp,maxiter=1E4)
75    psi_0=psi_0.reshape((-1,))
```

The calculation of the diagonal entropy density $s_d$ requires the eigensystem of the Hamiltonian $H_{\text{MBL}}(t_f)$ at the end of the ramp $t_f = (2v_f)^{-2}$. The entire spectrum and the corresponding eigenstates are obtained using the `hamiltonian` method `eigh`. For time-dependent Hamiltonians, `eigh` accepts the argument `time` to specify the time slice. Unless explicitly specified, `time=0.0` by default. Note that we re-set the ramp speed `v` to unity to calculate the correct eigensystem of the Hamiltonian at the end of the ramp, since $v$ a parameter of $H(t;v)$ (see line 68).

```
76    # calculate the eigensystem of the final MBL hamiltonian
77    E_final,V_final=H_MBL.eigh(time=(0.5/vs[-1]))
```

To calculate the entropies for each ramp speed, we use the helper function `_do_ramp` (defined below), which first evolves the initial state according to the $v$-dependent Hamiltonian $H_{\text{MBL}}(t;v)$ for a fixed ramp speed $v$. In line 78 we loop over the ramp speed vector `vs`. More importantly, however, the iteration index `v` carries the same name as the parameter in the drive function `ramp`. Thus, every time a new ramp speed is read off the vector `vs`, the external parameter `v` changes its value. Because `v` is a global variable, this change induces a change into the function `ramp` which, in turn, induces a change in the `dynamic` list. Thus, at the end of the day, a new member of the family of MBL Hamiltonians, $\{v : H_{\text{MBL}}(t;v)\}$, is picked and parsed to `_do_ramp` to do the time evolution with. Hence, we end up with a convenient and automatic way of generating the whole family $\{v : H_{\text{MBL}}(t;v)\}$, while having to calculate the operators in the Hamiltonian only once.

```
78    # evolve states and calculate entropies in MBL phase
79    run_MBL=[_do_ramp(psi_0,H_MBL,basis,v,E_final,V_final) for v in vs]
80    run_MBL=np.vstack(run_MBL).T
```

It remains to discuss the helper function `_do_ramp`. Its job is to evolve the initial state `psi_0` with the `hamiltonian` object `H` and to calculate the entropies at the end of the ramp.

```
100   ##### evolve state and evaluate entropies #####
101   def _do_ramp(psi_0,H,basis,v,E_final,V_final):
102       """
103       Auxiliary function to evolve the state and calculate the entropies after the
104       ramp.
105       --- arguments ---
106       psi_0: initial state
```

```
107      H: time-dependent Hamiltonian
108      basis: spin_basis_1d object containing the spin basis (required for Sent)
109      E_final, V_final: eigensystem of H(t_f) at the end of the ramp t_f=1/(2v)
110      """
```

Given a ramp speed `v`, we first determine the total ramp time `t_f`. Evolving a quantum state under any Hamiltonian H is easily done with the `hamiltonian` method `evolve`, see `line 114`. `evolve` requires the initial state `psi_0`, the starting time – here `0.0`, and a vector of times to return the evolved state at, but since we are only interested in the state at the final time – we pass the final time `t_f`. The `evolve` method has further interesting features which we discuss in Secs. 2.3 and 2.4.

```
111      # determine total ramp time
112      t_f = 0.5/v
113      # time-evolve state from time 0.0 to time t_f
114      psi = H.evolve(psi_0,0.0,t_f)
```

Once we have the state at the end of the ramp, we can obtain the entropies as follows. Calculating $s_{ent}$ is done using the `measurements` function `ent_entropy` which we imported in `line 3`. It requires the quantum state (here the pure state `psi`), and the `basis` the state is stored in[8]. Optionally, one can specify the site indices which define the subsystem retained after the partial trace using the argument `chain_subsys`. Note that `ent_entropy` returns a dictionary, in which the value of the entanglement entropy is stored under the key `"Sent"`. The function `ent_entropy` has a many further features, described in the documentation, see App. D.

```
115      # calculate entanglement entropy
116      subsys = range(basis.L//2) # define subsystem
117      Sent = ent_entropy(psi,basis,chain_subsys=subsys)["Sent"]
```

Similarly, there is a built-in function to calculate the diagonal entropy density $s_d$ of a state `psi` in a given basis (here `V_final`), called `diag_ensemble`. This function can calculate a variety of interesting quantities in the diagonal ensemble defined by the eigensystem arguments (here `E_final`, `V_final`). We again invite the interested reader to check out the documentation in App. D.

```
118      # calculate diagonal entropy in the basis of H(t_f)
119      S_d = diag_ensemble(basis.L,psi,E_final,V_final,Sd_Renyi=True)["Sd_pure"]
```

This concludes the definition of `_do_ramp`.

Back to the function `realization`, we have already seen how to obtain the entropies in the MBL phase. We now do the same thing in the delocalised ETH phase. The code is the same as the MBL one:

```
81       ###  ramp in ETH phase ###
82       v=1.0 # reset ramp speed to unity
83       # calculate the energy at infinite temperature for initial ETH hamiltonian
84       Emin,Emax=H_ETH.eigsh(time=0.0,**eigsh_args)
85       E_inf_temp=(Emax+Emin)/2.0
86       # calculate nearest eigenstate to energy at infinite temperature
87       E,psi_0=H_ETH.eigsh(time=0.0,k=1,sigma=E_inf_temp,maxiter=1E4)
88       psi_0=psi_0.reshape((-1,))
89       # calculate the eigensystem of the final ETH hamiltonian
90       E_final,V_final=H_ETH.eigh(time=(0.5/vs[-1]))
91       # evolve states and calculate entropies in ETH phase
92       run_ETH=[_do_ramp(psi_0,H_ETH,basis,v,E_final,V_final) for v in vs]
93       run_ETH=np.vstack(run_ETH).T # stack vertically elements of list run_ETH
```

---

[8]The `basis` is required since the subsystem may not share the same symmetries as the entire chain.

We can now display how long the single iteration took

```
95      # show time taken
96      print("realization {0}/{1} took {2:.2f} sec".format(real+1,n_real,time()-ti))
```

and conclude the definition of `realization`:

```
98      return run_MBL,run_ETH
```

Now that we have written the `realization` function, we can call it `n_real` times to produce the data. The easiest way of doing this is to loop over the disorder realisation, as shown in `lines 126-129`. However, a better to proceed makes use of the `joblib` package which can distribute simultaneous function calls over `n_job` Python processes, see `line 130`[9,10]. To learn more about how this works, we invite the readers to check the documentation of `joblib`. Having produced and extracted the entropy vs. ramp speed data, we are ready to perform the disorder average by taking the mean over all realisations, `lines 133-135`.

```
123  ##### produce data for n_real disorder realisations #####
124  # __name__ == '__main__' required to use joblib in Windows.
125  if __name__ == '__main__':
126      """
127      # alternative way without parallelisation
128      data = np.asarray([realization(vs,H_XXZ,basis,i) for i in range(n_real)])
129      """
130      data = np.asarray(Parallel(n_jobs=n_jobs)(delayed(realization)(vs,H_XXZ,basis
         ,i) for i in range(n_real)))
131      #
132      run_MBL,run_ETH = zip(*data) # extract MBL and data
133      # average over disorder
134      mean_MBL = np.mean(run_MBL,axis=0)
135      mean_ETH = np.mean(run_ETH,axis=0)
```

The complete code including the lines that produce Fig. 1 is available in Example Code 2.

## 2.3 Heating in periodically driven spin chains

*Physics Setup*—As a second example, we now show how one can easily study heating in the periodically-driven transverse-field Ising model with a parallel field [40–42]. This model is non-integrable even without the time-dependent driving protocol. The time-periodic Hamiltonian is defined as a two-step protocol as follows:

$$
H(t) = \begin{cases} J\sum_{j=0}^{L-1}\sigma_j^z\sigma_{j+1}^z + h\sum_{j=0}^{L-1}\sigma^z, & t \in [-T/4,\ T/4] \\ g\sum_{j=0}^{L-1}\sigma_j^x, & t \in [\ T/4,3T/4] \end{cases} \bmod T,
$$

$$
= \sum_{j=0}^{L-1}\frac{1}{2}\left(J\sigma_j^z\sigma_{j+1}^z + h\sigma^z + g\sigma_j^x\right) + \frac{1}{2}\text{sgn}[\cos\Omega t]\left(J\sigma_j^z\sigma_{j+1}^z + h\sigma^z - g\sigma_j^x\right). \quad (5)
$$

Unlike the previous example, here we consider a closed spin chain with a periodic boundary (i.e. a ring). The spin-spin interaction strength is denoted by $J$, the transverse field – by $g$, and the parallel field – by $h$. The period of the drive is $T$ and, although the periodic step protocol contains infinitely many Fourier harmonics, we shall refer to $\Omega = 2\pi/T$ as *the* frequency of the drive.

---

[9]The `if`-statement `if __name__ == "__main__"` in `line 125` is required by `joblib` to protect the parallel loop from recursively spawning python processes on Windows OS. For Linux/OS X it can safely be omitted.

[10]Because `joblib` spawns independent Python processes, the global variable `v` is not shared between them and so changing the value of `v` in one process will not effect the other processes.

Since the Hamiltonian is periodic, $H(t + T) = H(t)$, Floquet's theorem applies and postulates that the dynamics of the system at times $lT$, integer multiple of the driving period (a.k.a. stroboscopic times), is governed by the time-independent Floquet Hamiltonian $H_F$. In other words, the evolution operator is stroboscopically given by

$$U(lT) = \mathcal{T}_t \exp\left(-i \int_0^{lT} H(t)\mathrm{d}t\right) = \exp(-ilTH_F). \tag{6}$$

While the Floquet Hamiltonian for this system cannot be calculated analytically, a suitable approximation can be found at high drive frequencies by means of the van Vleck inverse-frequency expansion [22, 43]. However, this expansion is known to calculate the effective Floquet Hamiltonian $H_{\text{eff}}$ in a different basis than the original stroboscopic one:

$$H_F = \exp[-iK_{\text{eff}}(0)]H_{\text{eff}}\exp[iK_{\text{eff}}(0)],$$

which requires the additional calculation of the so-called Kick operator $K_{\text{eff}}(0)$ to 'rotate' to the original basis.

In the inverse-frequency expansion, we expand both $H_{\text{eff}}$ and $K_{\text{eff}}(0)$ in powers of the inverse frequency. Let us label these approximate objects by the superscript $^{(n)}$, suggesting that the corresponding operators are of order $\mathcal{O}(\Omega^{-n})$:

$$H_F = H_F^{(0)} + H_F^{(1)} + H_F^{(2)} + H_F^{(3)} + \mathcal{O}(\Omega^{-4}) = H_F^{(0+1+2+3)} + \mathcal{O}(\Omega^{-4}),$$
$$H_{\text{eff}} = H_{\text{eff}}^{(0)} + H_{\text{eff}}^{(1)} + H_{\text{eff}}^{(2)} + H_{\text{eff}}^{(3)} + \mathcal{O}(\Omega^{-4}),$$
$$K_{\text{eff}} = K_{\text{eff}}^{(0)} + K_{\text{eff}}^{(1)} + K_{\text{eff}}^{(2)} + K_{\text{eff}}^{(3)} + \mathcal{O}(\Omega^{-4}),$$

Using the short-hand notation one can show that, for this problem, all odd-order terms in the van Vleck expansion vanish [see App. G of Ref. [44]]

$$H_F^{(0+1+2+3)} = H_F^{(0+2)} \approx \mathrm{e}^{-iK_{\text{eff}}^{(2)}(0)}\left(H_{\text{eff}}^{(0)} + H_{\text{eff}}^{(2)}\right)\mathrm{e}^{+iK_{\text{eff}}^{(2)}(0)}, \tag{7}$$

while the first few even-order ones are given by

$$H_{\text{eff}}^{(0)} = \frac{1}{2}\sum_j J\sigma_j^z\sigma_{j+1}^z + h\sigma_j^z + g\sigma_j^x,$$

$$H_{\text{eff}}^{(2)} = -\frac{\pi^2}{12\Omega^2}\sum_j J^2 g\sigma_{j-1}^z\sigma_j^x\sigma_{j+1}^z + Jgh(\sigma_j^x\sigma_{j+1}^z + \sigma_j^z\sigma_{j+1}^x) + Jg^2(\sigma_j^y\sigma_{j+1}^y - \sigma_j^z\sigma_{j+1}^z)$$
$$+ \left(J^2 g + \frac{1}{2}h^2 g\right)\sigma_j^x + \frac{1}{2}hg^2\sigma_j^z,$$

$$K_{\text{eff}}^{(0)} = \mathbf{0},$$

$$K_{\text{eff}}^{(2)}(0) = \frac{\pi^2}{8\Omega^2}\sum_j Jg\left(\sigma_j^z\sigma_{j+1}^y + \sigma_j^y\sigma_{j+1}^z\right) + hg\sigma_j^y, \tag{8}$$

It was recently argued based on the aforementioned Floquet theorem that, in a closed periodically driven system, stroboscopic dynamics is sufficient to completely quantify heating [45], and we shall make use of this fact in our little study here. We choose as the initial state the ground state of the approximate Hamiltonian $H_F^{(0+1+2+3)}$ and denote it by $|\psi_i\rangle$:

$$|\psi_i\rangle = |\text{GS}(H_F^{(0+1+2+3)})\rangle. \tag{9}$$

Regimes of slow and fast heating can then be easily detected by looking at the energy density $\mathscr{E}$ absorbed by the system from the drive

$$\mathscr{E}(lT) = \frac{1}{L} \langle \psi_i | e^{ilTH_F} H_F^{(0+1+2)} e^{-ilTH_F} | \psi_i \rangle, \tag{10}$$

and the entanglement entropy of a subsystem. We call this subsystem A and define it to contain $L/2$ consecutive chain sites[11]:

$$s_{\text{ent}}(lT) = -\frac{1}{L_A} \text{tr}_A \big[ \rho_A(lT) \log \rho_A(lT) \big], \ \text{with} \ \rho_A(lT) = \text{tr}_{A^c} \big[ e^{-ilTH_F} | \psi_i \rangle \langle \psi_i | e^{ilTH_F} \big], \tag{11}$$

where the partial trace in the definition of the reduced density matrix (DM) $\rho_A$ is over the complement of A, denoted $A^c$, and $L_A = L/2$ denotes the length of subsystem A.

Since heating can be exponentially slow at high frequencies [46–49], one might be interested in calculating also the infinite-time quantities

$$\overline{\mathscr{E}} = \lim_{N \to \infty} \frac{1}{N} \sum_{l=0}^{N} \mathscr{E}(lT), \qquad \overline{s}_{\text{rdm}} = -\frac{1}{L_A} \text{tr}_A \big[ \overline{\rho}_A \log \overline{\rho}_A \big], \qquad s_d^F = -\frac{1}{L} \text{tr} \big[ \rho_d^F \log \rho_d^F \big], \tag{12}$$

where $\overline{\rho}_A$ is the infinite-time reduced DM of subsystem A, and $\rho_d^F$ is the DM of the Diagonal ensemble [50] in the exact Floquet basis $\{|n_F\rangle : H_F |n_F\rangle = E_F |n_F\rangle\}$:

$$\overline{\rho}_A = \lim_{N \to \infty} \frac{1}{N} \sum_{l=0}^{N} \rho_A(lT) = \text{tr}_{A^c} \big[ \rho_d^F \big], \qquad \rho_d^F = \sum_n |\langle \psi_i | n_F \rangle|^2 |n_F\rangle \langle n_F|.$$

We note in passing that in general $\overline{s}_{\text{rdm}} \neq \lim_{N \to \infty} N^{-1} \sum_{l=0}^{N} s_{\text{ent}}(lT)$ due to interference terms, although the two may happen to be close.

In Fig. 2 we show the time evolution of $\mathscr{E}(lT)$ and $s_{\text{ent}}(lT)$ as a function of the number of driving cycles $l$ for a given drive frequency, together with their infinite-time values.

*Code Analysis*—Let us now discuss the QuSpin code for this problem in detail. First we load the required classes, methods and functions required for the computation:

```
from quspin.operators import hamiltonian # Hamiltonians and operators
from quspin.basis import spin_basis_1d # Hilbert space spin basis
from quspin.tools.measurements import obs_vs_time, diag_ensemble # t_dep
    measurements
from quspin.tools.Floquet import Floquet, Floquet_t_vec # Floquet Hamiltonian
import numpy as np # generic math functions
```

After that, we define the model parameters:

```
L=14 # system size
J=1.0 # spin interaction
g=0.809 # transverse field
h=0.9045 # parallel field
Omega=4.5 # drive frequency
```

The time-periodic step drive of frequency `Omega` can easily be incorporated through the following function:

```
# define time-reversal symmetric periodic step drive
def drive(t,Omega):
    return np.sign(np.cos(Omega*t))
drive_args=[Omega]
```

[11]Since we use periodic boundaries, it does not matter which consecutive sites we choose. In fact, in QuSpin the user can choose any (possibly disconnected) subsystem to calculate the entanglement entropy and the reduced DM, see the documentation in App. D.

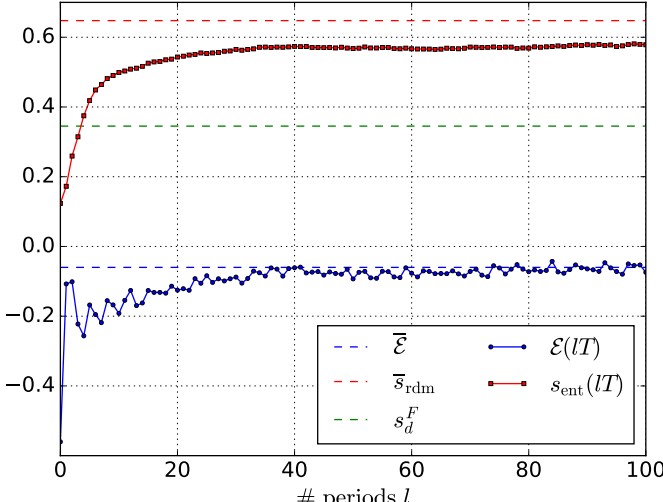

Figure 2: Stroboscopic dynamics of the energy density and entanglement entropy density (solid lines), together with their infinite-time values (dashed lines) in the periodically-driven TFIM in a parallel field. The parameters are $g/J = 0.809$, $h/J = 0.9045$, $\Omega/J = 4.5$, and $L = 14$.

Next, we define the basis, similar to the example in Sec. 2.2. One can convince oneself that the Hamiltonian in Eq. (5) possesses two symmetries at all times $t$ which are, therefore, also inherited by the Floquet Hamiltonian. These are translation invariance and parity (i.e. reflection w.r.t. the centre of the chain). To incorporate them, one needs to specify the desired block for each symmetry: `kblock=int` selects the many-body states of total momentum $2\pi/L*$`int`, while `pblock=±1` sets the parity sector. For all total momenta different from 0 and $\pi$, the translation operator does not commute with parity, in which case semi-momentum states producing a *real* Hamiltonian are the natural choice [51]. The optional argument `a=1` specifies the number of sites per unit cell[12].

```
19  # compute basis in the 0-total momentum and +1-parity sector
20  basis=spin_basis_1d(L=L,a=1,kblock=0,pblock=1)
```

The definition of the site-coupling lists proceeds similarly to the MBL example above. It is interesting to note how the periodic boundary condition is encoded in `line 25` using the modulo operator `%`. Compared to open boundaries, the PBC `J_nn` list now also has a total of L elements, as many as there are sites and bonds on the ring.

```
21  # define PBC site-coupling lists for operators
22  x_field_pos=[[+g,i] for i in range(L)]
23  x_field_neg=[[-g,i] for i in range(L)]
24  z_field=[[h,i]     for i in range(L)]
25  J_nn=[[J,i,(i+1)%L] for i in range(L)] # PBC
```

To program the full Hamiltonian $H(t)$, we use the second line of Eq. (5). The time-independent part is defined using the static operator list. For the time-dependent part, we need to pass the function `drive` and its arguments `drive_args`, defined in `lines 15–18`, to all operators the drive couples to. In fact, QuSpin is smart enough to automatically sum up all operators multiplied by the same time-dependent function in any dynamic list created. Note that since we are dealing with a Hamiltonian defined by Pauli matrices and not the spin-1/2 operators, we drop the optional argument `pauli` for the `hamiltonian` class, since by default it is set to `pauli=True`.

---

[12]For example if one has a staggered magnetic field, the unit cell has two sites and we need a=2.

```
26  # static and dynamic lists
27  static=[["zz",J_nn],["z",z_field],["x",x_field_pos]]
28  dynamic=[["zz",J_nn,drive,drive_args],
29          ["z",z_field,drive,drive_args],["x",x_field_neg,drive,drive_args]]
30  # compute Hamiltonians
31  H=0.5*hamiltonian(static,dynamic,dtype=np.float64,basis=basis)
```

The following lines define the approximate van Vleck Floquet Hamiltonian, cf. Eq. (8). Of particular interest is `line 37` where we define the site-coupling list for the three-spin operator `"zxz"`. Apart from the coupling `J**2*g`, we now need to specify the *three* site indices `i,(i+1)%L,(i+2)%L` for each of the operators `"zxz"`, respectively. In a similar fashion, one can define any multi-spin operator.

```
33  ##### set up second-order van Vleck Floquet Hamiltonian #####
34  # zeroth-order term
35  Heff_0=0.5*hamiltonian(static,[],dtype=np.float64,basis=basis)
36  # second-order term: site-coupling lists
37  Heff2_term_1=[[+J**2*g,i,(i+1)%L,(i+2)%L] for i in range(L)] # PBC
38  Heff2_term_2=[[+J*g*h, i,(i+1)%L] for i in range(L)] # PBC
39  Heff2_term_3=[[-J*g**2,i,(i+1)%L] for i in range(L)] # PBC
40  Heff2_term_4=[[+J**2*g+0.5*h**2*g,i] for i in range(L)]
41  Heff2_term_5=[[0.5*h*g**2,      i] for i in range(L)]
42  # define static list
43  Heff_static=[["zxz",Heff2_term_1],
44              ["xz",Heff2_term_2],["zx",Heff2_term_2],
45              ["yy",Heff2_term_3],["zz",Heff2_term_2],
46              ["x",Heff2_term_4],
47              ["z",Heff2_term_5]                          ]
48  # compute van Vleck Hamiltonian
49  Heff_2=hamiltonian(Heff_static,[],dtype=np.float64,basis=basis)
50  Heff_2*=-np.pi**2/(12.0*Omega**2)
51  # zeroth + second order van Vleck Floquet Hamiltonian
52  Heff_02=Heff_0+Heff_2
```

In order to rotate the state from the van Vleck to the stroboscopic (Floquet-Magnus) picture, we also have to calculate the kick operator at time $t = 0$. While the procedure is the same as above, note that $K_{\text{eff}}(0)$ has imaginary matrix elements, whence the variable `dtype=np.complex128` is used (in fact this is the default `dtype` optional argument that the `hamiltonian` class assumes if one does not pass this argument explicitly). If the user tries to force define a real-valued Hamiltonian which, however, has complex matrix elements, QuSpin will raise an error.

```
54  ##### set up second-order van Vleck Kick operator #####
55  Keff2_term_1=[[J*g,i,(i+1)%L] for i in range(L)] # PBC
56  Keff2_term_2=[[h*g,i] for i in range(L)]
57  # define static list
58  Keff_static=[["zy",Keff2_term_1],["yz",Keff2_term_1],["y",Keff2_term_2]]
59  Keff_02=hamiltonian(Keff_static,[],dtype=np.complex128,basis=basis)
60  Keff_02*=np.pi**2/(8.0*Omega**2)
```

Next, we need to find $H_F^{(0+2)} = \exp[-iK_{\text{eff}}^{(2)}(0)]H_{\text{eff}}^{(0+2)}\exp[iK_{\text{eff}}^{(2)}(0)]$. To this end, we make use of the `hamiltonian` class method `rotate_by` which conveniently provides a function for this purpose. By specifying the optional argument `generator=True`, `rotate_by` recognises the operator $B$ as a generator and defines a linear transformation to 'rotate' a `hamiltonian` object $A$ via $\exp(a^*B^\dagger)A\exp(aB)$ for any complex-valued number $a$. Although we do not make use of it directly here, it might also be useful for the user to become familiar with the documentation of the `exp_op` class which provides the matrix exponential, cf. App. D, and contains a variety of useful method functions. For instance, $\exp(zB)A$ can be obtained using

`exp_op(B,a=z).dot(A)`, while $A\exp(zB)$ is `A.dot(exp_op(B,a=z))`[13] for any complex number `z`.

```
62  ##### rotate Heff to stroboscopic basis #####
63  # e^{-1j*Keff_02} Heff_02 e^{+1j*Keff_02}
64  HF_02 = Heff_02.rotate_by(Keff_02,generator=True,a=1j)
```

Now that we have concluded the initialisation of the approximate Floquet Hamiltonian, it is time to discuss how to study the dynamics of the system. We start by defining a vector of times `t`, particularly suitable for the study of periodically driven systems. We initialise this time vector as an object of the `Floquet_t_vec` class. The arguments we need are the drive frequency `Omega`, the number of periods (here `100`), and the number of time points per period `len_T` (here set to 1). Once initialised, `t` has many useful attributes, such as the time values `t.vals`, the drive period `t.T`, the stroboscopic times `t.strobo.vals`, or their indices `t.strobo.inds`. The `Floquet_t_vec` class has further useful properties, described in the documentation in App. D.

```
66  ##### define time vector of stroboscopic times with 100 cycles #####
67  t=Floquet_t_vec(Omega,100,len_T=1) # t.vals=times, t.i=init. time, t.T=drive
        period
```

To calculate the exact stroboscopic Floquet Hamiltonian $H_F$, one can conveniently make use of the `Floquet` class. Currently, it supports three different ways of obtaining the Floquet Hamiltonian: (i) passing an arbitrary time-periodic `hamiltonian` object it will evolve each Fock state for one period to obtain the evolution operator $U(T)$. This calculation can be parallelised using the Python module `joblib`, activated by setting the optional argument `n_jobs`. (ii) one can pass a list of static `hamiltonian` objects, accompanied by a list of time steps to apply each of these Hamiltonians at. In this case, the `Floquet` class will make use of the matrix exponential to find $U(T)$. Instead, here we choose, (iii), to use a single dynamic `hamiltonian` object $H(t)$, accompanied by a list of times $\{t_i\}$ to evaluate it at, and a list of time steps $\{\delta t_i\}$ to compute the time-ordered matrix exponential as $\prod_i \exp(-iH(t_i)\delta t_i)$. The `Floquet` class calculates the quasienergies `EF` folded in the interval $[-\Omega/2, \Omega/2]$ by default. If required, the user may further request the set of Floquet states by setting `VF=True`, the Floquet Hamiltonian, `HF=True`, and/or the Floquet phases – `thetaF=True`. For more information on `Floquet_t_vec`, the user is advised to consult the package documentation, see App. D.

```
69  ##### calculate exact Floquet eigensystem #####
70  t_list=np.array([0.0,t.T/4.0,3.0*t.T/4.0])+np.finfo(float).eps # times to
        evaluate H
71  dt_list=np.array([t.T/4.0,t.T/2.0,t.T/4.0]) # time step durations to apply H for
72  Floq=Floquet({'H':H,'t_list':t_list,'dt_list':dt_list},VF=True) # call Floquet
        class
73  VF=Floq.VF # read off Floquet states
74  EF=Floq.EF # read off quasienergies
```

As discussed in the setup of the problem, we choose for the initial state the ground state[14] of the approximate Hamiltonian $H_F^{(0+2)}$. Following the discussion in Sec. 2.1, we use the `hamiltonian` class attribute `eigsh`[15].

```
76  ##### calculate initial state (GS of HF_02) and its energy
77  EF_02, psi_i = HF_02.eigsh(k=1,which="SA",maxiter=1E4)
78  psi_i = psi_i.reshape((-1,))
```

---

[13]One can also use the syntax `A.rdot(exp_op(a*B))` and `exp_op(z*B).rdot(A)`, respectively, for multiplication from the right.

[14]The approximate Floquet Hamiltonian is unfolded [38] and, thus, the ground state is well-defined.

[15]`which="SA"` tells `eigsh` to solve for the smallest algebraic eigenvalue.

Finally, we can calculate the time-dependence of the energy density $\mathscr{E}(t)$ and the entanglement entropy density $s_{\text{ent}}(t)$. This is done using the `measurements` function `obs_vs_time`. If one evolves with a constant Hamiltonian (which is effectively the case for stroboscopic time evolution), QuSpin offers two different but equivalent options, that we now discuss. (i) As a first required argument of `obs_vs_time` one passes a tuple `(psi_i,E,V)` with the initial state, the spectrum, and the eigenbasis of the Hamiltonian to do the evolution with. The second argument is the time vector (here `t.vals`), and the third one – a dictionary with the operator one would like to measure (here the approximate energy density `HF_02/L`, see line 83 below. If the observable is time-dependent, `obs_vs_time` will evaluate it at the appropriate times: $\langle\psi(t)|\mathscr{O}(t)|\psi(t)\rangle$. To obtain the entanglement entropy, `obs_vs_time` calls the `measurements` function `ent_entropy`, whose arguments are passed using the variable `Sent_args`. `ent_entropy` requires the `basis`, and optionally – the subsystem `chain_subsys` which would otherwise be set to the first L/2 sites of the chain. To learn more about how to obtain the reduced density matrix or other features of `ent_entropy`, consult the documentation, App. D.

```
80  ##### time-dependent measurements
81  # calculate measurements
82  Sent_args = {"basis":basis,"chain_subsys":[j for j in range(L//2)]}
83  #meas = obs_vs_time((psi_i,EF,VF),t.vals,{"E_time":HF_02/L},Sent_args=Sent_args)
```

The other way to calculate a time-dependent observable (ii) is more generic and works for arbitrary time-dependent Hamiltonians. It makes use of Schrödinger evolution to find the time-dependent state using the `evolve` method of the `hamiltonian` class. While we introduced `evolve` in Sec. 2.2, here we explain an important feature: if the optional argument `iterate=True` is passed, then QuSpin will not do the calculation of the state immediately; instead – it will create a generator object. This generator object will calculate the time dependent state one by one upon request. By doing so one can avoid the causal loop over the times `t.vals` to first find the state, and then looping once more over time to evaluate observables. The `evolve` method typically works for larger system sizes than the ones that allow full ED. One can then simply pass the generator `psi_t` into `obs_vs_time` instead of the initial tuple.

```
85  # alternative way by solving Schroedinger's eqn
86  psi_t = H.evolve(psi_i,t.i,t.vals,iterate=True,rtol=1E-9,atol=1E-9)
87  meas = obs_vs_time(psi_t,t.vals,{"E_time":HF_02/L},Sent_args=Sent_args)
```

The output of `obs_vs_time` is a dictionary. Extracting the energy density and entanglement entropy density values as a function of time, is as easy as:

```
89  # read off measurements
90  Energy_t = meas["E_time"]
91  Entropy_t = meas["Sent_time"]["Sent"]
```

Last, we compute the infinite-time value of the energy density, the entropy of the infinite-time reduced density matrix, as well as the Floquet diagonal entropy. They are, in fact, closely related to the expectation values of the Diagonal ensemble of the initial state in the Floquet basis [45]. The `measurements` tool contains the function `diag_ensemble` specifically designed for this purpose. The required arguments are the system size L, the initial state `psi_i`, as well as the Floquet spectrum `EF` and states `VF`. The optional arguments are packed in the auxiliary dictionary `DE_args`, and contain the observable `Obs`, the diagonal entropy `Sd_Renyi`, and the entanglement entropy of the reduced DM `Srdm_Renyi` with its arguments `Srdm_args`. The additional label `_Renyi` is used since in general one can also compute the Renyi entropy with parameter $\alpha$, if desired. The function `diag_ensemble` will automatically return the densities of the requested quantities, unless the flag `densities=False` is specified. It has more features which allow one to calculate the temporal and quantum fluctuations of an observable at

infinite times (i.e. in the Diagonal ensemble), and return the diagonal density matrix. Moreover, it can do additional averages of all diagonal ensemble quantities over a user-specified energy distribution, which may prove useful in calculating thermal expectations at infinite times, cf. App. D.

```
93  ##### calculate diagonal ensemble measurements
94  DE_args = {"Obs":HF_02,"Sd_Renyi":True,"Srdm_Renyi":True,"Srdm_args":Sent_args}
95  DE = diag_ensemble(L,psi_i,EF,VF,**DE_args)
96  Ed = DE["Obs_pure"]
97  Sd = DE["Sd_pure"]
98  Srdm=DE["Srdm_pure"]
```

The complete code including the lines that produce Fig. 2 is available in Example Code 3.

## 2.4 Quantised light-atom interactions in the semi-classical limit: recovering the periodically driven atom

*Physics Setup*—The last example we show deals with the quantisation of the (monochromatic) electromagnetic (EM) field. For the purpose of our study, we take a two-level atom (i.e. a single-site spin chain) and couple it to a single photon mode (i.e. a quantum harmonic oscillator). The Hamiltonian reads

$$H = \Omega a^\dagger a + \frac{A}{2}\frac{1}{\sqrt{N_{\mathrm{ph}}}}\left(a^\dagger + a\right)\sigma^x + \Delta\sigma^z, \tag{13}$$

where the operator $a^\dagger$ creates a photon in the mode, and the atom is modelled by a two-level system described by the Pauli spin operators $\sigma^{x,y,z}$. The photon frequency is $\Omega$, $N_{\mathrm{ph}}$ is the average number of photons in the mode, $A$ – the coupling between the EM field $E = \sqrt{N_{\mathrm{ph}}^{-1}}\left(a^\dagger + a\right)$ and the dipole operator $\sigma^x$, and $\Delta$ measures the energy difference between the two atomic states.

An interesting question to ask is under what conditions the atom can be described[16] by the time-periodic semi-classical Hamiltonian:

$$H_{\mathrm{sc}}(t) = A\cos\Omega t\ \sigma^x + \Delta\sigma^z. \tag{14}$$

Curiously, despite its simple form, one cannot solve in a closed form for the dynamics generated by the semi-classical Hamiltonian $H_{\mathrm{sc}}(t)$.

To address the above question, we prepare the system such that the atom is in its ground state, while we put the photon mode in a coherent state with mean number of photons $N_{\mathrm{ph}}$, as required to by the semi-classical regime [52]:

$$|\psi_i\rangle = |\mathrm{coh}(N_{\mathrm{ph}})\rangle|\downarrow\rangle. \tag{15}$$

We then calculate the exact dynamics generated by the spin-photon Hamiltonian $H$, measure the Pauli spin matrix $\sigma^z$ which represents the energy of the atom, $\sigma^x$ – the 'dipole' operator, and the photon number $n = a^\dagger a$:

$$\langle\mathcal{O}\rangle = \langle\psi_i|\mathrm{e}^{itH}\mathcal{O}\,\mathrm{e}^{-itH}|\psi_i\rangle, \qquad \mathcal{O} = n,\sigma^z,\sigma^y, \tag{16}$$

and compare these to the semi-classical expectation values

$$\langle\mathcal{O}\rangle_{\mathrm{sc}} = \langle\downarrow|\mathcal{T}_t\mathrm{e}^{i\int_0^t H_{\mathrm{sc}}(t')\mathrm{d}t'}\mathcal{O}\,\mathcal{T}_t\mathrm{e}^{-i\int_0^t H_{\mathrm{sc}}(t')\mathrm{d}t'}|\downarrow\rangle, \qquad \mathcal{O} = \sigma^z,\sigma^y. \tag{17}$$

---

[16]Strictly speaking the Hamiltonian $H_{\mathrm{sc}}(t)$ describes the spin dynamics in the rotating frame of the photon, defined by $a \to a\mathrm{e}^{-i\Omega t}$; however, all three observables of interest: $a^\dagger a$, and $\sigma^{y,z}$ are invariant under this transformation.

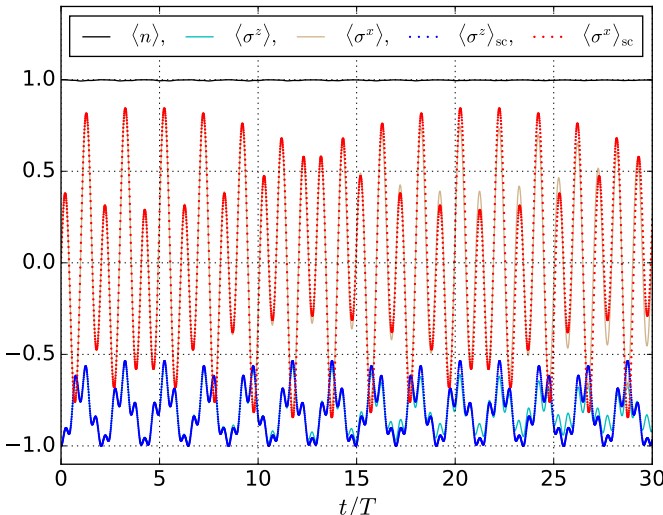

Figure 3: Emergent effective periodically driven dynamics in the semi-classical limit of the quantised light-atom interaction. The solid lines represent expectation values in the spin-photon basis, while dashed lines – the corresponding semi-classical quantities. The parameters are $A/\Delta = 1$, $\Omega/\Delta = 3.5$. The photon Hilbert space has a total number of $N_{\mathrm{ph,tot}} = 60$ states, and the mean number of photons in the initial coherent state is $N_{\mathrm{ph}} = 30$.

Figure 3 a shows a comparison between the quantum and the semi-classical time evolution of all observables $\mathcal{O}$ as defined above. As expected, we find a reasonable agreement with the deviation at longer times increasing, depending on the number of photons used, and the drive frequency.

*Code Analysis*—We used the following compact QuSpin code to produce these results. First we load the required classes, methods and functions to do the calculation:

```
1 from quspin.basis import spin_basis_1d,photon_basis # Hilbert space bases
2 from quspin.operators import hamiltonian # Hamiltonian and observables
3 from quspin.tools.measurements import obs_vs_time # t_dep measurements
4 from quspin.tools.Floquet import Floquet,Floquet_t_vec # Floquet Hamiltonian
5 from quspin.basis.photon import coherent_state # HO coherent state
6 import numpy as np # generic math functions
```

Next, we define the model parameters as follows:

```
8 ##### define model parameters #####
9 Nph_tot=60 # maximum photon occupation
10 Nph=Nph_tot/2 # mean number of photons in initial coherent state
11 Omega=3.5 # drive frequency
12 A=0.8 # spin-photon coupling strength (drive amplitude)
```

To set up the spin-photon Hamiltonian, we first build the site-coupling lists. The `ph_energy` list does not require the specification of a lattice site index, since the latter is not defined for the photon sector. The `at_energy` list, on the other hand, requires the input of the lattice site for the $\sigma^z$ operator: since we consider a single two-level system or, equivalently – a single-site chain, this index is `0`. The spin-photon coupling lists `absorb` and `emit` also require the site index which refers to the corresponding Pauli matrices: in this model – `0` again due to dimensional constraints.

```
16 # define operator site-coupling lists
17 ph_energy=[[Omega]] # photon energy
18 at_energy=[[Delta,0]] # atom energy
```

```
19 absorb=[[A/(2.0*np.sqrt(Nph)),0]] # absorption term
20 emit=[[A/(2.0*np.sqrt(Nph)),0]] # emission term
```

To build the static operator list, we use the | symbol in the operator string to distinguish the spin and photon operators: spin operators always come to the left of the |-symbol, while photon operators – to the right. For convenience, the identity operator I can be omitted, such that I|n is the same as |n, and z|I is equivalent to z|, respectively. The dynamic list is empty since the spin-photon Hamiltonian is time-independent.

```
21 # define static and dynamics lists
22 static=[["|n",ph_energy],["x|-",absorb],["x|+",emit],["z|",at_energy]]
23 dynamic=[]
```

To build the spin-photon basis, we call the function `photon_basis` and use `spin_basis_1d` as the first argument. We need to specify the number of spin lattice sites, and the total number of harmonic oscillator (a.k.a photon) states. Building the Hamiltonian works as in Sec. 2.2 and 2.3.

```
24 # compute atom-photon basis
25 basis=photon_basis(spin_basis_1d,L=1,Nph=Nph_tot)
26 # compute atom-photon Hamiltonian H
27 H=hamiltonian(static,dynamic,dtype=np.float64,basis=basis)
```

We now set up the time-periodic semi-classical Hamiltonian which is defined on the spin Hilbert space only; thus we use a `spin_basis_1d` basis object. We use the `dynamic_sc` list to define the time-dependence.

```
30 # define operators
31 dipole_op=[[A,0]]
32 # define periodic drive and its parameters
33 def drive(t,Omega):
34     return np.cos(Omega*t)
35 drive_args=[Omega]
36 # define semi-classical static and dynamic lists
37 static_sc=[["z",at_energy]]
38 dynamic_sc=[["x",dipole_op,drive,drive_args]]
39 # compute semi-classical basis
40 basis_sc=spin_basis_1d(L=1)
41 # compute semi-classical Hamiltonian H_{sc}(t)
42 H_sc=hamiltonian(static_sc,dynamic_sc,dtype=np.float64,basis=basis_sc)
```

Next, we define the initial state as a product state, see Eq. (15). Notice that in the QuSpin `spin_basis_1d` basis convention the state $|\downarrow\rangle = (1,0)^t$. This is because the spin basis states are coded using their bit representations and the state of all spins pointing down is assigned the integer 0. To define the oscillator (a.k.a. photon) coherent state with mean photon number $N_{ph}$, we use the function `coherent_state`: its first argument is the eigenvalue of the annihilation operator $a$, while the second argument is the total number of oscillator states[17].

```
45 # define atom ground state
46 psi_at_i=np.array([1.0,0.0]) # spin-down eigenstate of \sigma^z
47 # define photon coherent state with mean photon number Nph
48 psi_ph_i=coherent_state(np.sqrt(Nph),Nph_tot+1)
49 # compute atom-photon initial state as a tensor product
50 psi_i=np.kron(psi_at_i,psi_ph_i)
```

The next step is to define a vector of stroboscopic times, using the class `Floquet_t_vec`. Unlike in Sec. 2.3, here we are also interested in the non-stroboscopic times in between the

---

[17]Since the oscillator ground state is denoted by $|0\rangle$, the state $|N_{ph}\rangle$ is the $(N_{ph}+1)^{st}$ state of the oscillator basis, see line 48.

perfect periods $lT$. Thus, we omit the optional argument `len_T` making use of the default value set to `len_T=100`, meaning that there are now 100 time points within each period.

```
53 # define time vector over 30 driving cycles with 100 points per period
54 t=Floquet_t_vec(Omega,30) # t.i = initial time, t.T = driving period
```

We now time evolve the initial state both in the atom-photon, and the semi-classical cases using the `hamiltonian` class method `evolve`, as before. Once again, we define the solution `psi_t` as a generator expression using the optional argument `iterate=True`.

```
53 # evolve atom-photon state with Hamiltonian H
54 psi_t=H.evolve(psi_i,t.i,t.vals,iterate=True,rtol=1E-9,atol=1E-9)
55 # evolve atom GS with semi-classical Hamiltonian H_sc
56 psi_sc_t=H_sc.evolve(psi_at_i,t.i,t.vals,iterate=True,rtol=1E-9,atol=1E-9)
```

Last, we define the observables of interest, using the `hamiltonian` class with unit coupling constants. Since each observable represents a single operator, we refrain from defining operator lists and set up the observables in-line. Note that the main difference in defining the Pauli operators in the atom-photon and the semi-classical cases below (apart from the | notation), is the `basis` argument, defined in `lines 62-63`. The Python dictionaries `obs_args` and `obs_args_sc` represent another way of passing optional keyword arguments to the hamiltonian function. Here we also disable the automatic symmetry and hermiticity checks.

```
61 # define observables parameters
62 obs_args={"basis":basis,"check_herm":False,"check_symm":False}
63 obs_args_sc={"basis":basis_sc,"check_herm":False,"check_symm":False}
64 # in atom-photon Hilbert space
65 n=hamiltonian([["|n", [[1.0  ]] ]],[],dtype=np.float64,**obs_args)
66 sz=hamiltonian([["z|",[[1.0,0]] ]],[],dtype=np.float64,**obs_args)
67 sy=hamiltonian([["y|",  [[1.0,0]] ]],[],dtype=np.complex128,**obs_args)
68 # in the semi-classical Hilbert space
69 sz_sc=hamiltonian([["z",[[1.0,0]] ]],[],dtype=np.float64,**obs_args_sc)
70 sy_sc=hamiltonian([["y",[[1.0,0]] ]],[],dtype=np.complex128,**obs_args_sc)
```

Finally, we calculate the time-dependent expectation values using the `measurements` tool function `obs_vs_time`. Its arguments are the time-dependent state `psi_t`, the vector of times `t.vals`, and a dictionary of all observables of interest, and were discussed in Sec. 2.3. `obs_vs_time` returns a dictionary with all time-dependent expectations stored under the same keys they were passed. They can be accessed as shown in lines 75 and 78, respectively.

```
73 # in atom-photon Hilbert space
74 Obs_t = obs_vs_time(psi_t,t.vals,{"n":n,"sz":sz,"sy":sy})
75 O_n, O_sz, O_sy = Obs_t["n"], Obs_t["sz"], Obs_t["sy"]
76 # in the semi-classical Hilbert space
77 Obs_sc_t = obs_vs_time(psi_sc_t,t.vals,{"sz_sc":sz_sc,"sy_sc":sy_sc})
78 O_sz_sc, O_sy_sc = Obs_sc_t["sz_sc"], Obs_sc_t["sy_sc"]
```

The complete code including the lines that produce Fig. 3 is available in Example Code 4.

# 3 Future perspectives for QuSpin

We have demonstrated that the QuSpin functionality allows the user to do many different kinds of ED calculations. In one spatial dimension, one also has the option of using a wide range of available symmetries. The reader might have noticed that, provided the study does not require the use of symmetries (e.g. a fully disordered 2D model), one can specify the site-coupling lists such as to build higher-dimensional Hamiltonians, using the `spin_basis_1d` class. Setting up

higher-dimensional Hamiltonians with symmetries is possible in limited cases, too, when they can be uniquely mapped to one-dimensional systems.

In addition to the features we have discussed in this paper, there are many other functions defined in QuSpin which are all listed in the Documentation (Appendix D). Some of the more interesting ones include the `tensor_basis` class which constructs a new basis object out of two other basis objects, thus implementing the tensor product. This can be employed, e.g., to study interacting ladders with hard-core bosons. Another class which is useful for state-of-the-art calculations is the `HamiltonianOperator` class. It does the matrix vector product without actually storing the matrix elements which significantly reduces the amount of memory needed to do this operation. This is particularly suited for diagonalising very large spin chains using `eigsh`, as it only requires on the order of a hundred calls of the matrix vector product to solve for a few eigenvalues and eigenvectors (for a specific example, see the documentation in App. D). A recent addition to the `tools` module is the `block_tools` module. This module contains a class which projects an initial state in the full Hilbert space to a set of user provided symmetry sectors and then evolves each block in parallel (possibly over many CPU core's) with a single function call. This is useful in cases where the intial state may not obey the given symmetries of the Hamiltonian used to evolve the state, for example when calculating non-equal time correlation functions. Finally the `hamiltonian` class is not just limited to matrices generated in our code from the operator strings. In general, this class also takes arbitrary matrices as inputs for both static and dynamic operators; therefore, one can use all of the packages functionality for any user-defined matrix.

We have set up the code to make it easily generalisable to different types of systems. We are currently working towards adding the one-dimensional symmetries for spinless and spinful fermions, but we are hoping to eventually add higher spins and bosons, too. Farther into the future we may implement a number of two dimensional lattices as well as their symmetries. We also welcome anyone who is interested in contributing to this project to reach out to the authors with any questions they may have about the package organisation. All modifications can be proposed through the pull request system on github.com.

We would much appreciate it if the users could report bugs using the issues forum in the QuSpin online repository.

# Acknowledgements

We would like to thank A. Iazzi, L. Pollet, M. Kolodrubetz, P. Mehta, M. Panday, P. Patil, A. Polkovnikov, A. Sandvik, D. Sels, and S. Vajna for various stimulating discussions and for providing comments on the draft. The authors are pleased to acknowledge that the computational work reported on in this paper was performed on the Shared Computing Cluster which is administered by Boston University's Research Computing Services. The authors also acknowledge the Research Computing Services group for providing consulting support which has contributed to the results reported within this paper. We would also like to thank Github for providing the online resources to help develop and maintain this project.

**Funding information**    This work was supported by NSF DMR-1410126, NSF DMR-1506340 and ARO W911NF1410540.

## A  Installation guide in a few steps

QuSpin is currently only being supported for Python 2.7 and Python 3.5 and so one must make sure to install this version of Python. We recommend the use of the free package manager Anaconda which installs Python and manages its packages. For a lighter installation, one can use miniconda.

### A.1  Mac OS X/Linux

To install Anaconda/miniconda all one has to do is execute the installation script with administrative privilege. To do this, open up the terminal and go to the folder containing the downloaded installation file and execute the following command:

```
$ sudo bash <installation_file>
```

You will be prompted to enter your password. Follow the prompts of the installation. We recommend that you allow the installer to prepend the installation directory to your PATH variable which will make sure this installation of Python will be called when executing a Python script in the terminal. If this is not done then you will have to do this manually in your bash profile file:

```
$ export PATH="path_to/anaconda/bin:$PATH"
```

**Installing via Anaconda.**—Once you have Anaconda/miniconda installed, all you have to do to install QuSpin is to execute the following command into the terminal:

```
$ conda install -c weinbe58 quspin
```

If asked to install new packages just say 'yes'. To keep the code up-to-date, just run this command regularly.

**Installing Manually.**—Installing the package manually is not recommended unless the above method failed. Note that you must have the Python packages NumPy, SciPy, and Joblib installed before installing QuSpin. Once all the prerequisite packages are installed, one can download the source code from github and then extract the code to whichever directory one desires. Open the terminal and go to the top level directory of the source code and execute:

```
$ python setup.py install --record install_file.txt
```

This will compile the source code and copy it to the installation directory of Python recording the installation location to `install_file.txt`. To update the code, you must first completely remove the current version installed and then install the new code. The `install_file.txt` can be used to remove the package by running:

```
$ cat install_file.txt | xargs rm -rf.
```

### A.2  Windows

To install Anaconda/miniconda on Windows, download the installer and execute it to install the program. Once Anaconda/miniconda is installed open the conda terminal and do one of the following to install the package:

**Installing via Anaconda.**—Once you have Anaconda/miniconda installed all you have to do to install QuSpin is to execute the following command into the terminal:

```
> conda install -c weinbe58 quspin
```

If asked to install new packages just say 'yes'. To update the code just run this command regularly.

**Installing Manually.**—Installing the package manually is not recommended unless the above method failed. Note that you must have NumPy, SciPy, and Joblib installed before installing QuSpin. Once all the prerequisite packages are installed, one can download the source code from github and then extract the code to whichever directory one desires. Open the terminal and go to the top level directory of the source code and then execute:

```
> python setup.py install --record install_file.txt
```

This will compile the source code and copy it to the installation directory of Python and record the installation location to `install_file.txt`. To update the code you must first completely remove the current version installed and then install the new code.

# B Basic use of command line to run Python

In this appendix we will review how to use the command line for Windows and OS X/Linux to navigate your computer's folders/directories and run the Python scripts.

## B.1 Mac OS X/Linux

Some basic commands:

- change directory:
  ```
  $ cd < path_to_directory >
  ```

- list files in current directory:
  ```
  $ ls
  ```

  list files in another directory:
  ```
  $ ls < path_to_directory >
  ```

- make new directory:
  ```
  $ mkdir <path>/< directory_name >
  ```

- copy file:
  ```
  $ cp < path >/< file_name > < new_path >/< new_file_name >
  ```

- move file or change file name:
  ```
  $ mv < path >/< file_name > < new_path >/< new_file_name >
  ```

- remove file:
  ```
  $ rm < path_to_file >/< file_name >
  ```

Unix also has an autocomplete feature if one hits the TAB key. It will complete a word or stop when it matches more than one file/folder name. The current directory is denoted by "." and the directory above is "..". Now, to execute a Python script all one has to do is open your terminal and navigate to the directory which contains the python script. To execute the script just use the following command:

```
$ python script.py
```

It's that simple!

## B.2 Windows

Some basic commands:

- change directory:
  ```
  > cd < path_to_directory >
  ```

- list files in current directory:
  ```
  > dir
  ```

  list files in another directory:
  ```
  > dir < path_to_directory >
  ```

- make new directory:
  ```
  > mkdir <path>\< directory_name >
  ```

- copy file:
  ```
  > copy < path >\< file_name > < new_path >\< new_file_name >
  ```

- move file or change file name:
  ```
  > move < path >\< file_name > < new_path >\< new_file_name >
  ```

- remove file:
  ```
  > erase < path >\< file_name >
  ```

Windows also has a autocomplete feature using the TAB key but instead of stopping when there multiple files/folders with the same name, it will complete it with the first file alphabetically. The current directory is denoted by "." and the directory above is "..".

## B.3 Execute Python script (any operating system)

To execute a Python script all one has to do is open up a terminal and navigate to the directory which contains the Python script. Python can be recognised by the extension `.py`. To execute the script just use the following command:

```
python script.py
```

It's that simple!

## C   Complete example codes

In this appendix, we give the complete Python scripts for the four examples discussed in Sec. 2.
In case the reader has trouble with the TAB spaces when copying from the code environments
below, the scripts can be downloaded from github at:

https://github.com/weinbe58/QuSpin/tree/master/examples

QuSpin *Example Code* 1: Exact Diagonalisation of the XXZ Model

```python
from quspin.operators import hamiltonian # Hamiltonians and operators
from quspin.basis import spin_basis_1d # Hilbert space spin basis
import numpy as np # generic math functions
#
##### define model parameters #####
L=12 # system size
Jxy=np.sqrt(2.0) # xy interaction
Jzz_0=1.0 # zz interaction
hz=1.0/np.sqrt(3.0) # z external field
#
##### set up Heisenberg Hamiltonian in an enternal z-field #####
# compute spin-1/2 basis
basis = spin_basis_1d(L,pauli=False)
basis = spin_basis_1d(L,pauli=False,Nup=L//2) # zero magnetisation sector
basis = spin_basis_1d(L,pauli=False,Nup=L//2,pblock=1) # and positive parity
    sector
# define operators with OBC using site-coupling lists
J_zz = [[Jzz_0,i,i+1] for i in range(L-1)] # OBC
J_xy = [[Jxy/2.0,i,i+1] for i in range(L-1)] # OBC
h_z=[[hz,i] for i in range(L-1)]
# static and dynamic lists
static = [["+-",J_xy],["-+",J_xy],["zz",J_zz]]
dynamic=[]
# compute the time-dependent Heisenberg Hamiltonian
H_XXZ = hamiltonian(static,dynamic,basis=basis,dtype=np.float64)
#
##### various exact diagonalisation routines #####
# calculate entire spectrum only
E=H_XXZ.eigvalsh()
# calculate full eigensystem
E,V=H_XXZ.eigh()
# calculate minimum and maximum energy only
Emin,Emax=H_XXZ.eigsh(k=2,which="BE",maxiter=1E4,return_eigenvectors=False)
# calculate the eigenstate closest to energy E_star
E_star = 0.0
E,psi_0=H_XXZ.eigsh(k=1,sigma=E_star,maxiter=1E4)
psi_0=psi_0.reshape((-1,))
```

QuSpin *Example Code* 2: Adiabatic Control of Parameters in MBL Phases

```python
1  from quspin.operators import hamiltonian # Hamiltonians and operators
2  from quspin.basis import spin_basis_1d # Hilbert space spin basis
3  from quspin.tools.measurements import ent_entropy, diag_ensemble # entropies
4  from numpy.random import ranf,seed # pseudo random numbers
5  from joblib import delayed,Parallel # parallelisation
6  import numpy as np # generic math functions
7  from time import time # timing package
8  #
9  ##### define simulation parameters #####
10 n_real=20 # number of disorder realisations
11 n_jobs=2 # number of spawned processes used for parallelisation
12 #
13 ##### define model parameters #####
14 L=10 # system size
15 Jxy=1.0 # xy interaction
16 Jzz_0=1.0 # zz interaction at time t=0
17 h_MBL=3.9 # MBL disorder strength
18 h_ETH=0.1 # delocalised disorder strength
19 vs=np.logspace(-2.0,0.0,num=20,base=10) # log_2-spaced vector of ramp speeds
20 #
21 ##### set up Heisenberg Hamiltonian with linearly varying zz-interaction #####
22 # define linear ramp function
23 v = 1.0 # declare ramp speed variable
24 def ramp(t):
25     return (0.5 + v*t)
26 ramp_args=[]
27 # compute basis in the 0-total magnetisation sector (requires L even)
28 basis = spin_basis_1d(L,Nup=L//2,pauli=False)
29 # define operators with OBC using site-coupling lists
30 J_zz = [[Jzz_0,i,i+1] for i in range(L-1)] # OBC
31 J_xy = [[Jxy/2.0,i,i+1] for i in range(L-1)] # OBC
32 # static and dynamic lists
33 static = [["+-",J_xy],["-+",J_xy]]
34 dynamic =[["zz",J_zz,ramp,ramp_args]]
35 # compute the time-dependent Heisenberg Hamiltonian
36 H_XXZ = hamiltonian(static,dynamic,basis=basis,dtype=np.float64)
37 #
38 ##### calculate diagonal and entanglement entropies #####
39 def realization(vs,H_XXZ,basis,real):
40     """
41     This function computes the entropies for a single disorder realisation.
42     --- arguments ---
43     vs: vector of ramp speeds
44     H_XXZ: time-dep. Heisenberg Hamiltonian with driven zz-interactions
45     basis: spin_basis_1d object containing the spin basis
46     n_real: number of disorder realisations; used only for timing
47     """
48     ti = time() # get start time
49     #
50     global v # declare ramp speed v a global variable
51     #
52     seed() # the random number needs to be seeded for each parallel process
53     #
54     # draw random field uniformly from [-1.0,1.0] for each lattice site
55     unscaled_fields=-1+2*ranf((basis.L,))
56     # define z-field operator site-coupling list
```

```python
     h_z=[[unscaled_fields[i],i] for i in range(basis.L)]
     # static list
     disorder_field = [["z",h_z]]
     # compute disordered z-field Hamiltonian
     no_checks={"check_herm":False,"check_pcon":False,"check_symm":False}
     Hz=hamiltonian(disorder_field,[],basis=basis,dtype=np.float64,**no_checks)
     # compute the MBL and ETH Hamiltonians for the same disorder realisation
     H_MBL=H_XXZ+h_MBL*Hz
     H_ETH=H_XXZ+h_ETH*Hz
     #
     ### ramp in MBL phase ###
     v=1.0 # reset ramp speed to unity
     # calculate the energy at infinite temperature for initial MBL Hamiltonian
     eigsh_args={"k":2,"which":"BE","maxiter":1E4,"return_eigenvectors":False}
     Emin,Emax=H_MBL.eigsh(time=0.0,**eigsh_args)
     E_inf_temp=(Emax+Emin)/2.0
     # calculate nearest eigenstate to energy at infinite temperature
     E,psi_0=H_MBL.eigsh(time=0.0,k=1,sigma=E_inf_temp,maxiter=1E4)
     psi_0=psi_0.reshape((-1,))
     # calculate the eigensystem of the final MBL hamiltonian
     E_final,V_final=H_MBL.eigh(time=(0.5/vs[-1]))
     # evolve states and calculate entropies in MBL phase
     run_MBL=[_do_ramp(psi_0,H_MBL,basis,v,E_final,V_final) for v in vs]
     run_MBL=np.vstack(run_MBL).T
     #
     ###  ramp in ETH phase ###
     v=1.0 # reset ramp speed to unity
     # calculate the energy at infinite temperature for initial ETH hamiltonian
     Emin,Emax=H_ETH.eigsh(time=0.0,**eigsh_args)
     E_inf_temp=(Emax+Emin)/2.0
     # calculate nearest eigenstate to energy at infinite temperature
     E,psi_0=H_ETH.eigsh(time=0.0,k=1,sigma=E_inf_temp,maxiter=1E4)
     psi_0=psi_0.reshape((-1,))
     # calculate the eigensystem of the final ETH hamiltonian
     E_final,V_final=H_ETH.eigh(time=(0.5/vs[-1]))
     # evolve states and calculate entropies in ETH phase
     run_ETH=[_do_ramp(psi_0,H_ETH,basis,v,E_final,V_final) for v in vs]
     run_ETH=np.vstack(run_ETH).T # stack vertically elements of list run_ETH
     # show time taken
     print("realization {0}/{1} took {2:.2f} sec".format(real+1,n_real,time()-ti))
     #
     return run_MBL,run_ETH
#
##### evolve state and evaluate entropies #####
def _do_ramp(psi_0,H,basis,v,E_final,V_final):
     """
     Auxiliary function to evolve the state and calculate the entropies after the
     ramp.
     --- arguments ---
     psi_0: initial state
     H: time-dependent Hamiltonian
     basis: spin_basis_1d object containing the spin basis (required for Sent)
     E_final, V_final: eigensystem of H(t_f) at the end of the ramp t_f=1/(2v)
     """
     # determine total ramp time
     t_f = 0.5/v
     # time-evolve state from time 0.0 to time t_f
```

```python
114        psi = H.evolve(psi_0,0.0,t_f)
115        # calculate entanglement entropy
116        subsys = range(basis.L//2) # define subsystem
117        Sent = ent_entropy(psi,basis,chain_subsys=subsys)["Sent"]
118        # calculate diagonal entropy in the basis of H(t_f)
119        S_d = diag_ensemble(basis.L,psi,E_final,V_final,Sd_Renyi=True)["Sd_pure"]
120        #
121        return np.asarray([S_d,Sent])
122 #
123 ##### produce data for n_real disorder realisations #####
124 # __name__ == '__main__' required to use joblib in Windows.
125 if __name__ == '__main__':
126     """
127     # alternative way without parallelisation
128     data = np.asarray([realization(vs,H_XXZ,basis,i) for i in range(n_real)])
129     """
130     data = np.asarray(Parallel(n_jobs=n_jobs)(delayed(realization)(vs,H_XXZ,basis
           ,i) for i in range(n_real)))
131     #
132     run_MBL,run_ETH = zip(*data) # extract MBL and data
133     # average over disorder
134     mean_MBL = np.mean(run_MBL,axis=0)
135     mean_ETH = np.mean(run_ETH,axis=0)
136     #
137     ##### plot results #####
138     import matplotlib.pyplot as plt
139     ### MBL plot ###
140     fig, pltarr1 = plt.subplots(2,sharex=True) # define subplot panel
141     # subplot 1: diag enetropy vs ramp speed
142     pltarr1[0].plot(vs,mean_MBL[0],label="MBL",marker=".",color="blue") # plot
           data
143     pltarr1[0].set_ylabel("$s_d(t_f)$",fontsize=22) # label y-axis
144     pltarr1[0].set_xlabel("$v/J_{zz}(0)$",fontsize=22) # label x-axis
145     pltarr1[0].set_xscale("log") # set log scale on x-axis
146     pltarr1[0].grid(True,which='both') # plot grid
147     pltarr1[0].tick_params(labelsize=16)
148     # subplot 2: entanglement entropy vs ramp speed
149     pltarr1[1].plot(vs,mean_MBL[1],marker=".",color="blue") # plot data
150     pltarr1[1].set_ylabel("$s_\mathrm{ent}(t_f)$",fontsize=22) # label y-axis
151     pltarr1[1].set_xlabel("$v/J_{zz}(0)$",fontsize=22) # label x-axis
152     pltarr1[1].set_xscale("log") # set log scale on x-axis
153     pltarr1[1].grid(True,which='both') # plot grid
154     pltarr1[1].tick_params(labelsize=16)
155     # save figure
156     fig.savefig('example1_MBL.pdf', bbox_inches='tight')
157     #
158     ### ETH plot ###
159     fig, pltarr2 = plt.subplots(2,sharex=True) # define subplot panel
160     # subplot 1: diag enetropy vs ramp speed
161     pltarr2[0].plot(vs,mean_ETH[0],marker=".",color="green") # plot data
162     pltarr2[0].set_ylabel("$s_d(t_f)$",fontsize=22) # label y-axis
163     pltarr2[0].set_xlabel("$v/J_{zz}(0)$",fontsize=22) # label x-axis
164     pltarr2[0].set_xscale("log") # set log scale on x-axis
165     pltarr2[0].grid(True,which='both') # plot grid
166     pltarr2[0].tick_params(labelsize=16)
167     # subplot 2: entanglement entropy vs ramp speed
168     pltarr2[1].plot(vs,mean_ETH[1],marker=".",color="green") # plot data
```

```
169     pltarr2[1].set_ylabel("$s_\mathrm{ent}(t_f)$",fontsize=22) # label y-axis
170     pltarr2[1].set_xlabel("$v/J_{zz}(0)$",fontsize=22) # label x-axis
171     pltarr2[1].set_xscale("log") # set log scale on x-axis
172     pltarr2[1].grid(True,which='both') # plot grid
173     pltarr2[1].tick_params(labelsize=16)
174     # save figure
175     fig.savefig('example1_ETH.pdf', bbox_inches='tight')
176     #
177     plt.show() # show plots
```

QuSpin *Example Code* 3: Heating in Periodically Driven Spin Chains

```python
1  from quspin.operators import hamiltonian # Hamiltonians and operators
2  from quspin.basis import spin_basis_1d # Hilbert space spin basis
3  from quspin.tools.measurements import obs_vs_time, diag_ensemble # t_dep
       measurements
4  from quspin.tools.Floquet import Floquet, Floquet_t_vec # Floquet Hamiltonian
5  import numpy as np # generic math functions
6  #
7  ##### define model parameters #####
8  L=14 # system size
9  J=1.0 # spin interaction
10 g=0.809 # transverse field
11 h=0.9045 # parallel field
12 Omega=4.5 # drive frequency
13 #
14 ##### set up alternating Hamiltonians #####
15 # define time-reversal symmetric periodic step drive
16 def drive(t,Omega):
17     return np.sign(np.cos(Omega*t))
18 drive_args=[Omega]
19 # compute basis in the 0-total momentum and +1-parity sector
20 basis=spin_basis_1d(L=L,a=1,kblock=0,pblock=1)
21 # define PBC site-coupling lists for operators
22 x_field_pos=[[+g,i] for i in range(L)]
23 x_field_neg=[[-g,i] for i in range(L)]
24 z_field=[[h,i]      for i in range(L)]
25 J_nn=[[J,i,(i+1)%L] for i in range(L)] # PBC
26 # static and dynamic lists
27 static=[["zz",J_nn],["z",z_field],["x",x_field_pos]]
28 dynamic=[["zz",J_nn,drive,drive_args],
29         ["z",z_field,drive,drive_args],["x",x_field_neg,drive,drive_args]]
30 # compute Hamiltonians
31 H=0.5*hamiltonian(static,dynamic,dtype=np.float64,basis=basis)
32 #
33 ##### set up second-order van Vleck Floquet Hamiltonian #####
34 # zeroth-order term
35 Heff_0=0.5*hamiltonian(static,[],dtype=np.float64,basis=basis)
36 # second-order term: site-coupling lists
37 Heff2_term_1=[[+J**2*g,i,(i+1)%L,(i+2)%L] for i in range(L)] # PBC
38 Heff2_term_2=[[+J*g*h, i,(i+1)%L] for i in range(L)] # PBC
39 Heff2_term_3=[[-J*g**2,i,(i+1)%L] for i in range(L)] # PBC
40 Heff2_term_4=[[+J**2*g+0.5*h**2*g,i] for i in range(L)]
41 Heff2_term_5=[[0.5*h*g**2,        i] for i in range(L)]
42 # define static list
43 Heff_static=[["zxz",Heff2_term_1],
44             ["xz",Heff2_term_2],["zx",Heff2_term_2],
45             ["yy",Heff2_term_3],["zz",Heff2_term_2],
46             ["x",Heff2_term_4],
47             ["z",Heff2_term_5]                          ]
48 # compute van Vleck Hamiltonian
49 Heff_2=hamiltonian(Heff_static,[],dtype=np.float64,basis=basis)
50 Heff_2*=-np.pi**2/(12.0*Omega**2)
51 # zeroth + second order van Vleck Floquet Hamiltonian
52 Heff_02=Heff_0+Heff_2
53 #
54 ##### set up second-order van Vleck Kick operator #####
55 Keff2_term_1=[[J*g,i,(i+1)%L] for i in range(L)] # PBC
```

```python
Keff2_term_2=[[h*g,i] for i in range(L)]
# define static list
Keff_static=[["zy",Keff2_term_1],["yz",Keff2_term_1],["y",Keff2_term_2]]
Keff_02=hamiltonian(Keff_static,[],dtype=np.complex128,basis=basis)
Keff_02*=np.pi**2/(8.0*Omega**2)
#
##### rotate Heff to stroboscopic basis #####
# e^{-1j*Keff_02} Heff_02 e^{+1j*Keff_02}
HF_02 = Heff_02.rotate_by(Keff_02,generator=True,a=1j)
#
##### define time vector of stroboscopic times with 100 cycles #####
t=Floquet_t_vec(Omega,100,len_T=1) # t.vals=times, t.i=init. time, t.T=drive
    period
#
##### calculate exact Floquet eigensystem #####
t_list=np.array([0.0,t.T/4.0,3.0*t.T/4.0])+np.finfo(float).eps # times to
    evaluate H
dt_list=np.array([t.T/4.0,t.T/2.0,t.T/4.0]) # time step durations to apply H for
Floq=Floquet({'H':H,'t_list':t_list,'dt_list':dt_list},VF=True) # call Floquet
    class
VF=Floq.VF # read off Floquet states
EF=Floq.EF # read off quasienergies
#
##### calculate initial state (GS of HF_02) and its energy
EF_02, psi_i = HF_02.eigsh(k=1,which="SA",maxiter=1E4)
psi_i = psi_i.reshape((-1,))
#
##### time-dependent measurements
# calculate measurements
Sent_args = {"basis":basis,"chain_subsys":[j for j in range(L//2)]}
#meas = obs_vs_time((psi_i,EF,VF),t.vals,{"E_time":HF_02/L},Sent_args=Sent_args)
#"""
# alternative way by solving Schroedinger's eqn
psi_t = H.evolve(psi_i,t.i,t.vals,iterate=True,rtol=1E-9,atol=1E-9)
meas = obs_vs_time(psi_t,t.vals,{"E_time":HF_02/L},Sent_args=Sent_args)
#"""
# read off measurements
Energy_t = meas["E_time"]
Entropy_t = meas["Sent_time"]["Sent"]
#
##### calculate diagonal ensemble measurements
DE_args = {"Obs":HF_02,"Sd_Renyi":True,"Srdm_Renyi":True,"Srdm_args":Sent_args}
DE = diag_ensemble(L,psi_i,EF,VF,**DE_args)
Ed = DE["Obs_pure"]
Sd = DE["Sd_pure"]
Srdm=DE["Srdm_pure"]
#
##### plot results #####
import matplotlib.pyplot as plt
import pylab
# define legend labels
str_E_t = "$\\mathcal{E}(lT)$"
str_Sent_t = "$s_\mathrm{ent}(lT)$"
str_Ed = "$\\overline{\mathcal{E}}$"
str_Srdm = "$\\overline{s}_\mathrm{rdm}$"
str_Sd = "$s_d^F$"
# plot infinite-time data
```

```python
110  fig = plt.figure()
111  plt.plot(t.vals/t.T,Ed*np.ones(t.vals.shape),"b--",linewidth=1,label=str_Ed)
112  plt.plot(t.vals/t.T,Srdm*np.ones(t.vals.shape),"r--",linewidth=1,label=str_Srdm)
113  plt.plot(t.vals/t.T,Sd*np.ones(t.vals.shape),"g--",linewidth=1,label=str_Sd)
114  # plot time-dependent data
115  plt.plot(t.vals/t.T,Energy_t,"b-o",linewidth=1,label=str_E_t,markersize=3.0)
116  plt.plot(t.vals/t.T,Entropy_t,"r-s",linewidth=1,label=str_Sent_t,markersize=3.0)
117  # label axes
118  plt.xlabel("$\\#\ \\mathrm{periods}\\ l$",fontsize=18)
119  # set y axis limits
120  plt.ylim([-0.6,0.7])
121  # display legend
122  plt.legend(loc="lower right",ncol=2,fontsize=18)
123  # update axis font size
124  plt.tick_params(labelsize=16)
125  # turn on grid
126  plt.grid(True)
127  # save figure
128  fig.savefig('example2.pdf', bbox_inches='tight')
129  # show plot
130  plt.show()
```

QuSpin *Example Code* 4: Quantised Light-Atom Interactions in the Semi-classical Limit

```python
from quspin.basis import spin_basis_1d,photon_basis # Hilbert space bases
from quspin.operators import hamiltonian # Hamiltonian and observables
from quspin.tools.measurements import obs_vs_time # t_dep measurements
from quspin.tools.Floquet import Floquet,Floquet_t_vec # Floquet Hamiltonian
from quspin.basis.photon import coherent_state # HO coherent state
import numpy as np # generic math functions
#
##### define model parameters #####
Nph_tot=60 # maximum photon occupation
Nph=Nph_tot/2 # mean number of photons in initial coherent state
Omega=3.5 # drive frequency
A=0.8 # spin-photon coupling strength (drive amplitude)
Delta=1.0 # difference between atom energy levels
#
##### set up photon-atom Hamiltonian #####
# define operator site-coupling lists
ph_energy=[[Omega]] # photon energy
at_energy=[[Delta,0]] # atom energy
absorb=[[A/(2.0*np.sqrt(Nph)),0]] # absorption term
emit=[[A/(2.0*np.sqrt(Nph)),0]] # emission term
# define static and dynamics lists
static=[["|n",ph_energy],["x|-",absorb],["x|+",emit],["z|",at_energy]]
dynamic=[]
# compute atom-photon basis
basis=photon_basis(spin_basis_1d,L=1,Nph=Nph_tot)
# compute atom-photon Hamiltonian H
H=hamiltonian(static,dynamic,dtype=np.float64,basis=basis)
#
##### set up semi-classical Hamiltonian #####
# define operators
dipole_op=[[A,0]]
# define periodic drive and its parameters
def drive(t,Omega):
    return np.cos(Omega*t)
drive_args=[Omega]
# define semi-classical static and dynamic lists
static_sc=[["z",at_energy]]
dynamic_sc=[["x",dipole_op,drive,drive_args]]
# compute semi-classical basis
basis_sc=spin_basis_1d(L=1)
# compute semi-classical Hamiltonian H_{sc}(t)
H_sc=hamiltonian(static_sc,dynamic_sc,dtype=np.float64,basis=basis_sc)
#
##### define initial state #####
# define atom ground state
psi_at_i=np.array([1.0,0.0]) # spin-down eigenstate of \sigma^z
# define photon coherent state with mean photon number Nph
psi_ph_i=coherent_state(np.sqrt(Nph),Nph_tot+1)
# compute atom-photon initial state as a tensor product
psi_i=np.kron(psi_at_i,psi_ph_i)
#
##### calculate time evolution #####
# define time vector over 30 driving cycles with 100 points per period
t=Floquet_t_vec(Omega,30) # t.i = initial time, t.T = driving period
# evolve atom-photon state with Hamiltonian H
psi_t=H.evolve(psi_i,t.i,t.vals,iterate=True,rtol=1E-9,atol=1E-9)
```

```python
57  # evolve atom GS with semi-classical Hamiltonian H_sc
58  psi_sc_t=H_sc.evolve(psi_at_i,t.i,t.vals,iterate=True,rtol=1E-9,atol=1E-9)
59  #
60  ##### define observables #####
61  # define observables parameters
62  obs_args={"basis":basis,"check_herm":False,"check_symm":False}
63  obs_args_sc={"basis":basis_sc,"check_herm":False,"check_symm":False}
64  # in atom-photon Hilbert space
65  n=hamiltonian([["|n", [[1.0  ]] ]],[],dtype=np.float64,**obs_args)
66  sz=hamiltonian([["z|",[[1.0,0]] ]],[],dtype=np.float64,**obs_args)
67  sy=hamiltonian([["y|",  [[1.0,0]] ]],[],dtype=np.complex128,**obs_args)
68  # in the semi-classical Hilbert space
69  sz_sc=hamiltonian([["z",[[1.0,0]] ]],[],dtype=np.float64,**obs_args_sc)
70  sy_sc=hamiltonian([["y",[[1.0,0]] ]],[],dtype=np.complex128,**obs_args_sc)
71  #
72  ##### calculate expectation values #####
73  # in atom-photon Hilbert space
74  Obs_t = obs_vs_time(psi_t,t.vals,{"n":n,"sz":sz,"sy":sy})
75  O_n, O_sz, O_sy = Obs_t["n"], Obs_t["sz"], Obs_t["sy"]
76  # in the semi-classical Hilbert space
77  Obs_sc_t = obs_vs_time(psi_sc_t,t.vals,{"sz_sc":sz_sc,"sy_sc":sy_sc})
78  O_sz_sc, O_sy_sc = Obs_sc_t["sz_sc"], Obs_sc_t["sy_sc"]
79  ##### plot results #####
80  import matplotlib.pyplot as plt
81  import pylab
82  # define legend labels
83  str_n = "$\\langle n\\rangle,$"
84  str_z = "$\\langle\\sigma^z\\rangle,$"
85  str_x = "$\\langle\\sigma^x\\rangle,$"
86  str_z_sc = "$\\langle\\sigma^z\\rangle_\\mathrm{sc},$"
87  str_x_sc = "$\\langle\\sigma^x\\rangle_\\mathrm{sc}$"
88  # plot spin-photon data
89  fig = plt.figure()
90  plt.plot(t.vals/t.T,O_n/Nph,"k",linewidth=1,label=str_n)
91  plt.plot(t.vals/t.T,O_sz,"c",linewidth=1,label=str_z)
92  plt.plot(t.vals/t.T,O_sy,"tan",linewidth=1,label=str_x)
93  # plot semi-classical data
94  plt.plot(t.vals/t.T,O_sz_sc,"b.",marker=".",markersize=1.8,label=str_z_sc)
95  plt.plot(t.vals/t.T,O_sy_sc,"r.",marker=".",markersize=2.0,label=str_x_sc)
96  # label axes
97  plt.xlabel("$t/T$",fontsize=18)
98  # set y axis limits
99  plt.ylim([-1.1,1.4])
100 # display legend horizontally
101 plt.legend(loc="upper right",ncol=5,columnspacing=0.6,numpoints=4)
102 # update axis font size
103 plt.tick_params(labelsize=16)
104 # turn on grid
105 plt.grid(True)
106 # save figure
107 fig.savefig('example3.pdf', bbox_inches='tight')
108 # show plot
109 plt.show()
```

# D  Package documentation

In QuSpin quantum many-body operators are represented as matrices. The computation of these matrices are done through custom code written in Cython. Cython is an optimizing static compiler which takes code written in a syntax similar to Python, and compiles it into a highly efficient C/C++ shared library. These libraries are then easily interfaced with Python, but can run orders of magnitude faster than pure Python code [53]. The matrices are stored in a sparse matrix format using the sparse matrix library of SciPy [24]. This allows QuSpin to easily interface with mature Python packages, such as NumPy, SciPy, any many others. These packages provide reliable state-of-the-art tools for scientific computation as well as support from the Python community to regularly improve and update them [24, 54–56]. Moreover, we have included specific functionality in QuSpin which uses NumPy and SciPy to do many desired calculations common to ED studies, while making sure the user only has to call a few NumPy or SciPy functions directly. The complete up-to-date documentation for the package is available online under: https://github.com/weinbe58/QuSpin/#quspin.

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
