# Peer review of "QuSpin: a Python Package for Dynamics and Exact Diagonalisation of Quantum Many Body Systems part I: spin chains"

_SciPost Physics, doi:SciPost Phys. 2, 003 (2017)_

## Round 2 · Referee Report · Anonymous · 2016-11-29

Strengths

1. Versatile Python package for exact diagonalization and (imaginary) time evolution of quantum spin chains
2. The authors provide their code as an open-source package to the community
3. The paper is well written, and the code examples nicely illustrate what can be done and how to use the package
4. Simple user interface and a clear installation guide

Weaknesses

1. Currently limited support for models with higher spins and/or higher-dimensional lattices including symmetries

Report

In this article the authors present an open-source Python package called “QuSpin” for exact diagonalization and (imaginary) time evolution of quantum spin chains. The capabilities and user interface of the package are explained in detail by several illustrative examples, starting with a simple exact diagonalization of a spin chain, followed by a time-dependent problem (an adiabatic ramping of parameters in the many-body localized XXZ model). Further examples include the study of heating in the periodically-driven transverse-field Ising model in a parallel field, and time evolution of a two-level system coupled to a single photon mode. Finally, the authors mention current and possible future developments of their package, including the support of symmetries for fermionic, higher spin-, and bosonic systems, and higher-dimensional lattices.

This package provides a useful and powerful tool for the study of quantum spin systems, and I personally find it great that the authors make their code publicly available, which I believe will be useful for the community. Main advantages of the package include the support of arbitrary multi-spin operators and various symmetries (in 1D), a simple Python interface, as well as additional useful functions (e.g. to compute entanglement entropies). One weakness of the package is the lack of support of symmetries for higher-dimensional systems, and higher-spin, bosonic and fermionic systems, which currently limits the range of models that the package can be directly applied to (in an efficient way). But nevertheless, I believe that this package serves as a good starting point also for future developments (done by the authors themselves or by the community), also thanks to the detailed and up-to-date online documentation (the link is given in appendix D).

For these reasons I can recommend publication of this article in its present form.

Requested changes

Just two minor suggestions:
1) On p 5 I was wondering how to implement periodic boundary conditions (PBC) instead of open boundary conditions. Maybe it would be good to either add a footnote to briefly explain it (or at least add a comment at the end of the first paragraph of Sec. 2.1 that PBC will be discussed later in Sec. 2.3)

2) The authors may also want to mention the TNT tensor library in their introduction which can be used, e.g. for ground state and time-evolution calculations:
Tensor Network Theory Library, Beta Version 1.2.0 (2016), S. Al-Assam, S. R. Clark, D. Jaksch and TNT Development team, www.tensornetworktheory.org

  • validity: high
  • significance: good
  • originality: good
  • clarity: high
  • formatting: excellent
  • grammar: excellent

Author:  Phillip Weinberg  on 2016-12-05  [id 80]

(in reply to Report 1 on 2016-11-29)

Hello,

Thanks for taking the time to review this work!

I have no objections to these changes. I shall make the changes and update the arxiv draft today.

Best,

Phillip Weinberg

---

## Round 2 · Referee Report · Anonymous · 2016-12-7

Strengths

1. Freely available and rather flexible package for exact diagonalization calculations
2. The only ED package I know of that handles translation symmetries
3. Genuine open-source licence (3-clause BSD)
4. A useful set of examples, hopefully these can be expanded in the future

Weaknesses

1. Currently limited to spin-1/2 systems and bosons, in 1D (presumably it is possible to do non-structured systems, or other lattices, but without the use of space symmetries).
2. No non-abelian symmetries, eg dihedral symmetry of combined translation plus parity reflection.
3. Example code is distributed separately to the main package.

Report

This is a very well written paper, describing a very useful software package. I was able to install the package and run the example programs without much trouble. I will certainly recommend this package to my students and others, when needing an exact diagonalization code beyond "quick and dirty MATLAB".

The software package is already very useful, and I hope that the authors (or other people) continue the development. To this end, some documentation on github specifically for potential contributors to the code would be useful.

There is quite a bit of confusion in the academic community as to what "open source" actually is, and there are many examples of software that is often referred to as "open source", where it doesn't actually meet the commonly accepted definition (eg, https://opensource.org/osd). I think it would be a useful addition to the manuscript if the authors want to include a brief statement on what they consider to be "open source", and why they chose their license. The ALPS libraries, for example, are freely available and claim to be open source, but fail the "open source" definition since it contains a 'no commercial use' clause in the license.

The TNT tensor library (new in v3 of the paper) appears to be neither open source, nor freely available. While https://ccpforge.cse.rl.ac.uk/gf/project/tntlibrary/ states "GNU General Public License (GPL)", the page http://www.tensornetworktheory.org/documentation/a00028.html contains a very restrictive license, including "No part of the Software may be reproduced, modified, transmitted or transferred in any form or by any means, electronic or mechanical, without the express permission of the University. The permission of the University is not required if the said reproduction, modification, transmission or transference is done without financial return, the conditions of this Licence are imposed upon the receiver of the product, and all original and amended source code is included in any transmitted product. You may be held legally responsible for any copyright infringement that is caused or encouraged by your failure to abide by these terms and conditions.". The conditions on including "all original and amended source code" is rather troubling, and seems to be rather ambiguous. Moreover, the TNT library doesn't appear to be freely available; the 'Current Status' link says: "Currently availably for download is the beta version of the core pre-compiled library libtnt.a", but I wasn't able to find out how to download even a pre-compiled version. Given that this recently appeared in the new version of the manuscript, I think the authors should clarify the "open source" status of this library before including it in a list of freely available, open source software projects.

Requested changes

1. The authors should consider the comments above.
2. The abstract mentions 3 examples, but there are actually 4 examples provided. Most users are likely to want to start with a simple ground-state or full diagonalization and may be put off by the complicated examples. Explicitly mentioning the example 1 diagonalization of the XXZ chain would be useful.
3. It took a while to find out where to download the examples, I was expecting to see a link where the examples are first introduced in section 2.1. Please also consider including the examples in the main github repository, rather than on a separate site.

  • validity: high
  • significance: good
  • originality: high
  • clarity: top
  • formatting: excellent
  • grammar: excellent

Author:  Phillip Weinberg  on 2017-01-12  [id 86]

(in reply to Report 2 on 2016-12-07)

Hello,

I have updated the arXiv with the requested revisions. Note that I have also added some sentences in the outlook section as there has been some new features added to the package which I would like to advertise.

Thanks,

Phil

---

## Round 4 · Author Response

The changes advised by the referees all seemed perfectly reasonable so I tried to address all of them in this submission.

---

## Round 4 · List of Changes

Referee report 1:

1) I added a footnote on page 5 to address this.

2) Added the TNT library to the introduction.

Referee report 2:

1) Concerning the issue of open-source, I added a comment to the introduction in order to make a distinction between the different packages listed.

2) I added a sentence to the abstract which states that there is a first example for doing standard ED calculations.

3) This didn't have much to do with the manuscript but I did update the website for QuSpin so that the examples are on the home page as well as adding an explicit link in the appendix which the readers can follow to find them.

Changes made outside of referee reports:

There have been some updates to the package since the article was posted so I decided to add some of those extra features to the conclusion of the paper.

---

## Editorial Decision

published